# Global chromatin conformation differences in the *Drosophila* dosage compensated chromosome X

Koustav Pal [1], Mattia Forcato[2], Daniel Jost[3,4], Thomas Sexton[5,9], Cédric Vaillant[6], Elisa Salviato [1], Emilia Maria Cristina Mazza[7], Enrico Lugli [7], Giacomo Cavalli [5] & Francesco Ferrari [1,8]*

In *Drosophila melanogaster* the single male chromosome X undergoes an average twofold transcriptional upregulation for balancing the transcriptional output between sexes. Previous literature hypothesised that a global change in chromosome structure may accompany this process. However, recent studies based on Hi-C failed to detect these differences. Here we show that global conformational differences are specifically present in the male chromosome X and detectable using Hi-C data on sex-sorted embryos, as well as male and female cell lines, by leveraging custom data analysis solutions. We find the male chromosome X has more mid-/long-range interactions. We also identify differences at structural domain boundaries containing BEAF-32 in conjunction with CP190 or Chromator. Weakening of these domain boundaries in male chromosome X co-localizes with the binding of the dosage compensation complex and its co-factor CLAMP, reported to enhance chromatin accessibility. Together, our data strongly indicate that chromosome X dosage compensation affects global chromosome structure.

[1] IFOM, the FIRC Institute of Molecular Oncology, Via Adamello 16, 20139 Milan, Italy. [2] Department of Life Sciences, University of Modena and Reggio Emilia, Via G. Campi 287, 41125 Modena, Italy. [3] University of Grenoble Alpes, CNRS, CHU Grenoble Alpes, Grenoble INP, TIMC-IMAG, Grenoble, France. [4] Laboratory of Biology and Modelling of the Cell, University of Lyon, ENS de Lyon, University of Claude Bernard, CNRS UMR 5239, Inserm U1210, F-69007 Lyon, France. [5] IGH, Institute of Human Genetics, CNRS UPR1142, 141 rue de la Cardonille, 34090 Montpellier, France. [6] University of Lyon, ENS de Lyon, University of Claude Bernard, CNRS, Laboratoire de Physique, 46 allée d'Italie, 69007 Lyon, France. [7] Laboratory of Translational Immunology, Humanitas Clinical and Research Center, Via A. Manzoni 56, 20089 Rozzano, Milan, Italy. [8] Institute of Molecular Genetics, National Research Council, Via Abbiategrasso 207, 27100 Pavia, Italy. [9] Present address: Institute of Genetics and Molecular and Cellular Biology (IGBMC), 1 Rue Laurent Fries, 67404 Illkirch, France. *email: francesco.ferrari@ifom.eu

osage compensation (DC) is the process responsible for balancing transcriptional output of sex chromosomes in species with different karyotypes between sexes. Diverse modes of DC have evolved in various organisms employing different mechanisms of transcriptional regulation[1,2]. In *Drosophila melanogaster*, DC is achieved by the twofold upregulation of active genes on the single copy chromosome X (chrX) in males. The entire process is orchestrated by the male-specific lethal (MSL) complex, including proteins and non-coding RNAs, also known as the dosage compensation complex (DCC)[3]. The DCC makes first contact with chrX at specific sites known as chromatin entry sites (CES)[4] or high-affinity sites (HAS)[5,6], which have a recurrent GA-rich sequence motif. PionX sites (pioneering sites on the X), a subset of HAS, are first bound upon establishment of DC and are characterized by an extended conserved sequence motif[7] specifically enriched on chrX.

Two recent publications explored the role of chromatin 3D organization in *D. melanogaster* DC. Ramirez et al.[8] showed that HAS sites are arranged in close proximity in the three-dimensional (3D) space to facilitate the targeting of chrX by DCC. Whereas Schauer et al.[9] showed that, after binding, the DCC spreads in *cis* along chrX from the initial binding sites to all of the active genes, aided by the 3D chromatin conformation around PionX sites. The DCC eventually deposits a specific activating chromatin mark (histone 4 lysine 16 acetylation—H4K16ac) over the body of transcribed genes[10].

Chromosome conformation capture techniques were instrumental towards these findings[11]. These techniques have progressively revealed a highly compartmentalized 3D structure of the genome. In particular, Hi-C opened the possibility to characterize physical proximity of any pair of genomic loci. This allowed the identification of higher order genome compartments distinguished by active or inactive chromatin[12]. Higher resolution data allow characterizing structural domains at a finer scale by identifying sub-megabase size domains, often termed topologically associating domains (TADs)[13–15]. TADs generally correlate with active and inactive chromatin states and various architectural proteins are usually found at TAD borders. While large-scale TAD structures are a general feature across Eukarya[16–18] there are differences in the players involved. BEAF-32 and CP190 are especially enriched at TAD borders in *D. melanogaster*[13], whereas CTCF and cohesin-mediated loops are the predominant feature of TAD borders in mammalian genomes[19,20]. Evidence in literature suggests that structural domains can be directly predicted from transcriptional states[16]. Nevertheless, TAD structure can be conserved across cell types even when the activity status within the region is different[21]. Changes, if any, are observed locally in chromatin interactions[22].

Studies on other model organisms have revealed an interplay between DC and 3D compartmentalization of the genome. In placental mammals, DC involves the silencing of one copy of chromosome X (chrX) in females, and is one of the most studied examples of epigenetic regulation[23]. This process is characterized by a complete reorganization of the dosage compensated chrX which is condensed into a highly compact structure[24,25]. In *Caenorhabditis elegans*, instead, DC reduces transcriptional output by an ~50% in the two copies of chrX in the hermaphrodite sex[26]. In this case, both copies display a distinct chromatin structure, with stronger insulation between structural domains, which is different from what is observed in autosomes[27,28].

In the dosage compensated *D. melanogaster* chrX, a global change in chromatin organization was also hypothesized, primarily on the basis of FISH experiments showing 3D co-localization of CES in male but not in female cells[29]. However, two recent publications based on Hi-C experiments could not detect large-scale conformational changes in male vs female

chrX[8,9]. However, the average twofold increase in transcription involved in *D. melanogaster* DC can be expected to yield a smaller change in chrX structure as compared with other chromosome-wide coordinated regulation of transcription, such as mammalian DC. In the latter case, chrX inactivation is characterized by marked condensation. Furthermore, the Hi-C data analysis methods adopted in these studies are established procedures, yet not specifically designed for the challenges of dealing with a single chromosome copy number difference.

In our study, we examine the 3D chromatin organization of dosage compensated chrX in *D. melanogaster* using high-resolution Hi-C data. First, we show that dosage compensated chrX engages in more mid- and long-range contacts, yet following a random pattern compatible with a globally more accessible structure. This observation is confirmed at multiple levels and across datasets, independent of underlying biases due to copy number differences. We also present evidence indicating the single copy male chrX as a more flexible polymer, partly due to the absence of homologous chromosome pairing. Then we report that male vs female differences in structural domains are primarily located on chrX, and that weakening of insulation at structural domain borders is associated to DC mechanisms. More specifically, weakening domain borders co-localize with DCC binding sites as well as active genes. These boundaries are also associated to pairs of canonical insulators (BEAF-32 along with CP190 or Chromator) participating in long-range interactions in both sexes. Near these boundaries, differential binding signal is observed for CLAMP, a DCC co-factor known to be involved in organizing chromatin architecture by enhancing accessibility. Finally, the long-range interactions between such boundaries near active genes are observed to be higher in male than in females.

In this work, the development of tailored Hi-C data analysis procedures was instrumental to effectively detect global and local differences in the interaction profiles and structural domains in dosage compensated chrX, which are robust and reproducible across datasets.

## Results

**Global differences in the interaction profiles for chrX.** To verify if the dosage compensated chrX in the male fly has a different 3D structure, we used high-resolution Hi-C data generated by simplified Hi-C protocol[13], containing more than a billion read pairs for the male and female sex-sorted embryos at 16–18 h developmental stage (GSE94115) (Supplementary Table 1). We first investigated the difference in the global interaction pattern between male and female chrX. When considering the $\log_2$ ratio between the independently normalized Hi-C data for chrX, we noticed a relative increase in the male over female Hi-C signal ratio at mid-/long-range distances (500 kb–1 Mb), with many points in the contact matrix showing a signal of relatively higher magnitude in the male chrX (Fig. 1a).

To confirm the observation and to quantify this pattern, we examined the Hi-C interaction frequency decay as a function of distance in male[8] and female[30] cell lines in addition to sex-sorted embryos, considering each autosomal chromosome arm and chrX independently (Fig. 1b). We observed that in the log–log plot of interaction frequencies vs distance, the chrX line is the only one different between male and female samples within the same dataset, i.e. considering sex-sorted or cell lines data, respectively. Namely, as distances increases, the chrX lines slowly drifts away from the autosome ones. This same pattern was not observed in the female samples. This suggested a slower interaction frequency decay with distance, which would indeed result in relatively higher interaction frequencies in the mid-/long-range distances. The pairwise differences (deltas; Δ) between the interaction decay

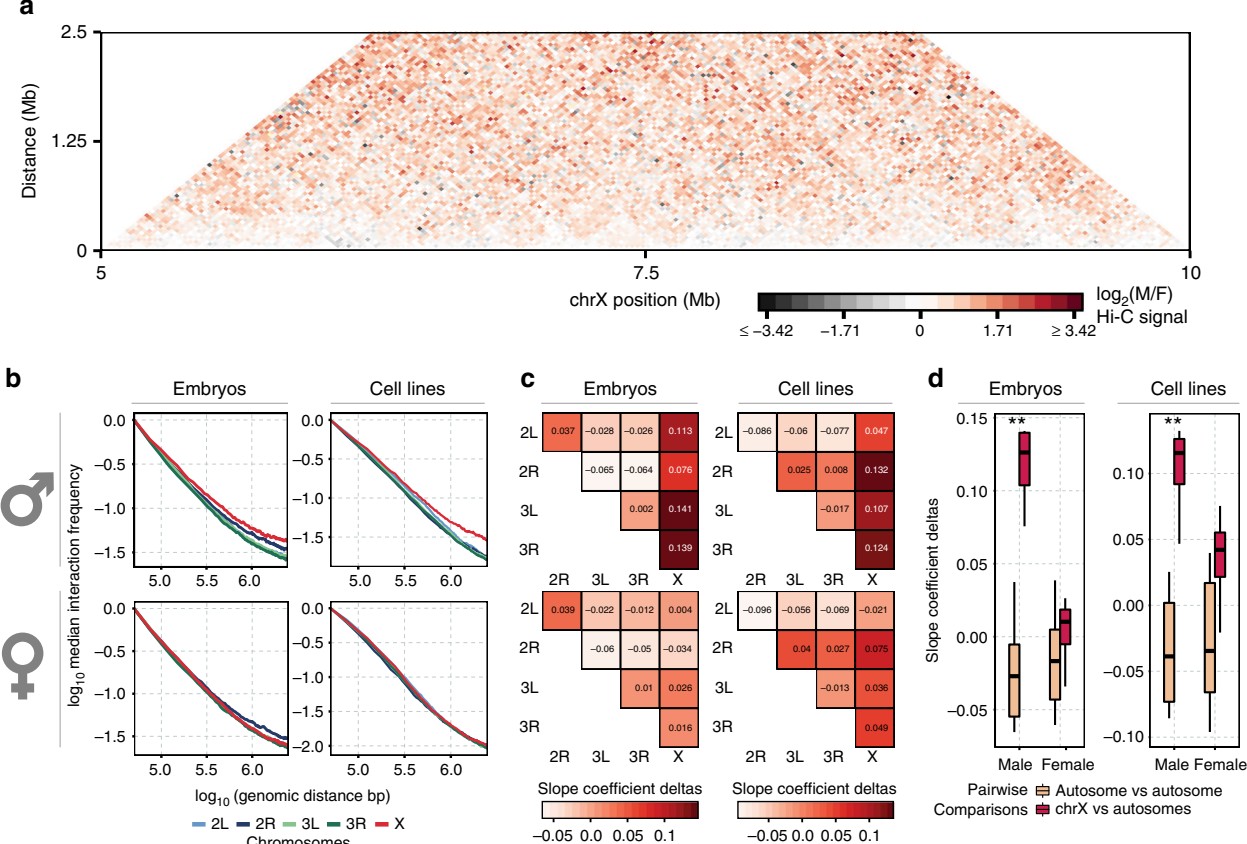

**Fig. 1** Higher interaction frequency at mid-/long-range distances in male chrX. **a** Male over female $\log_2$ ratio of normalized (chromosome-wise ICE) Hi-C matrices of contact frequencies binned at 25 kb resolution is shown for a representative 5 Mb region of chrX. **b** Interaction frequency decay with distance is shown in a log–log plot. Normalized (genome-wide ICE) Hi-C data binned at 25 kb were converted to contact frequencies as described in Methods for both sex-sorted embryos (left column) and cell lines (right column) datasets, for male (clone-8, upper row) and female (Kc167, lower row) samples. For any given genomic distance ($\log_{10}$ genomic distance on $x$-axis) the median $\log_{10}$ Hi-C interaction frequency is reported ($y$-axis). Distances ranging from 50 kb (2 bins distance in the 25 kb bins matrix) to 2.5 Mb are shown. **c** Heatmap showing the pairwise difference (deltas) of the rate of Hi-C signal decay (slope coefficient). The slope coefficients were computed considering distances <400 kb ($\leq 5.6$ in $\log_{10}$ distance). The colour scale and the numeric values report the differences between chromosomes indicated in the horizontal vs vertical axes labels. Layout as in (**b**) panel (male samples in upper row, female samples in lower row). **d** Boxplot of slope coefficient deltas for rate of Hi-C decay grouped by autosome pairwise comparisons (the first three columns in panel (**c**) heatmaps) and chrX comparisons to autosomes (the fourth column in panel (**c**) heatmaps) in male and female embryos (left) or cell lines (right). The difference in rate of decay between autosomes and chrX is highly significant in male samples only (** marks Wilcoxon test $p$-value < 0.01). For each box the median is marked as horizontal line, the boxes mark the interquartile range (IQR), the whiskers extend up to 1.5 IQR.

slope coefficients for each chromosome (see Methods) shows that chrX consistently has a less negative slope in male (average difference 0.11), but not in female samples (Fig. 1c). This observation is concordant across replicates (Supplementary Fig. 1a) and not dependent on the lower male chrX coverage, as expected due to copy number differences between sexes (Supplementary Fig. 1b—see Methods).

The reported difference is also robust to variations of several analysis parameters, including three normalization procedures: chromosome by chromosome ICE normalization (chromosome-wise ICE), genome-wide ICE and hicpipe-based explicit biases modelling[31,32] (Supplementary Fig. 2a). For each normalization we analysed the data with or without an alternative probabilistic transformation of the interaction frequency[33] (Supplementary Fig. 2b, see Methods). Finally, we confirmed that the observed differences are significant by comparing either the slope coefficients deltas (Wilcoxon test $p$-value < 0.01) (Fig. 1c) or the cumulative density functions of contact frequencies, which are independent of parameters estimation on the decay curves (Kuiper's statistic—see Methods) (Supplementary Fig. 1c). Thus, we can conclude that chrX interaction profile shows consistent

differences across Hi-C data for sex-sorted embryos and independent Hi-C datasets for cell lines, including clone-8 (male) and Kc167 (female) cell lines (Fig. 1b–d)[8,30].

Note that this male-specific interaction decay difference is not consistent in the S2 male cell line[8]. This is not surprising as S2 cells are notoriously affected by several copy number and structural variations[34,35]. These anomalies are bound to confound the genome-wide estimation of Hi-C interactions decay with distance. Furthermore, the tetraploid status of these cells, which carry two X chromosomes, might alter the general contact frequency decay behaviour.

**Interaction pattern shows that male chrX is more accessible.** We aimed to clarify if the higher interaction signal at mid- and long-range distances reflects a difference in specific point inter-actions. A first approach could be based on comparing the significant point interactions identified in the male and female Hi-C data matrices. However, as we observed in our recent study[36], all algorithms for detecting point interactions in Hi-C show a strong dependency between the number of called interactions and the

coverage (number of reads). As the single copy male chrX yields relatively fewer reads than the dual copy female chrX, this is expected to introduce a bias in the number of point interactions. To overcome this limitation, we devised a non-parametric approach to compare the pattern of interactions (see Methods). This procedure is based on selecting the highest scoring pairwise interactions by applying a threshold on quantiles computed independently for each diagonal of intra-chromosomal normalized Hi-C data matrices (Fig. 2a). In our approach, the resulting top-scoring interactions are by definition a fixed number of data points. Thus, they should not be considered as significant point interactions per se. Instead, we focus on assessing if they are differentially distributed across samples and chromosomes. The distribution of top-scoring interactions in male vs female autosome Hi-C maps is mostly conserved, yielding an almost symmetrical plot when reported side by side (Fig. 2b, left panel). Conversely, some parts of the chrX Hi-C map are symmetrical as well, while others are more different. Specifically, male chrX seems to have a less clustered pattern of top-scoring interactions (Fig. 2b, right panel).

To objectively assess these differences genome-wide, we quantified the fraction of clustered top-scoring interactions (see Methods). We observe that the difference in clustered data points in female vs male is larger for chrX than for autosomes, with male chrX consistently showing less clustered top-scoring interactions (Fig. 2c). This finding is robust to several parameter variations, including three normalization techniques (chromosome-wise ICE, genome-wide ICE and hicpipe-based explicit biases modelling)[31,32], and a wide range of percentile thresholds on top-scoring interactions and their maximum distance (Supplementary Fig. 3a, c). This observation holds true for our sex-sorted male and female Hi-C data, as well as independent Hi-C datasets for cell lines including Kc167 (female), clone-8 (male) and S2 (male) (Supplementary Fig. 3b). We also considered if the less clustered distribution of top-scoring interactions in the single copy male chrX may be due to lower coverage yielding noisier data. To further clarify this point, we repeated the analysis on Hi-C contact matrices obtained by downsampling read pairs for each female chromosome to the read count of the corresponding male chromosome (see Methods). The analysis of clustered top-scoring interactions confirms the same pattern, which is still present after reads downsampling (Supplementary Fig. 3a—downsampling) (Supplementary Data 1).

Thus, even if the male chrX shows more signal in mid- and long-range distances compared with the female (Fig. 1), the top-scoring interactions in the same distance range are more randomly distributed (Fig. 2). Taken together these results are compatible with a scenario where the dosage compensated chrX is globally more accessible, thus prone to yield non-specific and random ligation events, yet resulting in detectable differences in the Hi-C interactions profile.

A more accessible chrX is expected to yield more inter-chromosomal Hi-C contacts. Thus, we assessed the total inter-chromosomal (trans) contacts, defined as read pairs with either end mapped in different chromosomes, compared with the total intra-chromosomal (cis) contacts. We noted that trans-contacts are relatively more frequent specifically on male chrX, as confirmed both in sex-sorted embryos and cell lines datasets (Fig. 2d, and Supplementary Fig. 3d). This is compatible with a globally more accessible male chrX. However, we reasoned that this might also be due to a technicality of Hi-C data analysis, as trans-contacts between homologous chromosomes are indistinguishable from intra-chromosomal contacts. As male chrX is the only chromosome without a homologous one in our dataset, the relatively higher number of trans-contacts might just be due to the absence of confounding contacts between homologues.

To rule out this possibility, we examined the distribution of trans-contacts across chromosomes and verified that in sex-sorted embryos trans-contacts originating from each single male autosome are specifically enriched in chrX (Fig. 2e). This pattern is not compatible with a confounding effect due to the single copy of chrX and can only be explained by chrX indeed engaging in more trans-contacts with all other chromosomes. Conversely this pattern is not observed in female sex-sorted embryos (Fig. 2e). We must also note that this male-specific pattern is only partially confirmed in the cell lines datasets (Supplementary Fig. 3e).

Overall these observations are compatible with a scenario where the dosage compensated chrX presents a globally more open structure prone to random contacts detectable by Hi-C.

It is worth remarking that in D. melanogaster, like in other Dipteran insects, homologous chromosomes are paired throughout the cell cycle, i.e. in close physical proximity, although the exact molecular mechanisms are not completely characterized yet[37]. Previous literature estimated the Hi-C reads originating from homologous pairs to be rare in D. melanogaster, thus negligible for the global chromatin conformation profile[13]. Nevertheless, since we are specifically focusing on the dosage compensated male chrX, we asked whether the lack of a physically proximal homologous chromosome may affect the distribution of Hi-C contacts. Indeed, the Hi-C read pairs originating from inter-chromosomal (trans) contacts between homologues are indistinguishable from intra-chromosomal (cis) contacts, thus potentially confounding the estimation of cis-contacts. As the male chrX is the only chromosome without a paired homologue, we asked if this male chrX peculiarity may affect the observed differences in the interaction decay profile. To this concern, recent independent studies presenting haplotype-resolved Hi-C data show that the ratio of inter-homologous interactions vs cis-interactions increases at mid-/long-range distances[38,39]. Thus, the lack of a homologous chromosome may result in less mid-/long-range contacts being detected, i.e. the opposite of the trend observed in our results. This may suggest that the higher interaction frequency at mid-/long-range distance observed in the unpaired male chrX, compared with autosomes (Fig. 1b), is not a bias due to lack of pairing. However, the net effect on the relative contact frequency may not be so straightforward due to the consequent redistribution of inter-homologous read pairs, that could become either cis- or trans-read pairs in the absence of the homologous.

Thus, to definitely disentangle the contribution of homologous chromosomes pairing to the interaction decay profile, we applied polymer folding simulation techniques. Abed et al.[39] reported that the level of pairing is correlated to the local chromatin state, which in turn is known to affect the pattern of Hi-C contacts[16,40–42]. Therefore, we applied a polymer folding simulation model accounting for chromatin states annotations in the D. melanogaster genome as well as distribution of homologous tight pairing parameters in line with Abed et al.[39] estimations (see Methods). We observe that the lack of homologous pairing results in a globally less negative slope of contact frequency decay (Supplementary Fig. 4a and d). Even if the tight pairing is enriched in active chromatin, the shift in the interaction decay is equally observed in all chromatin states specifically on the unpaired chrX (Supplementary Fig. 4b). These results indicate that the lack of pairing may allow more flexibility in the chromatin fibre so as to allow more mid-/long-range contacts to occur. Interestingly, when stratifying the interactions by chromatin state in experimental data on sex-sorted embryos, the interaction decay plot shows a clear difference for the interactions originating from the active state, with less marked decay observed in all chromosomes

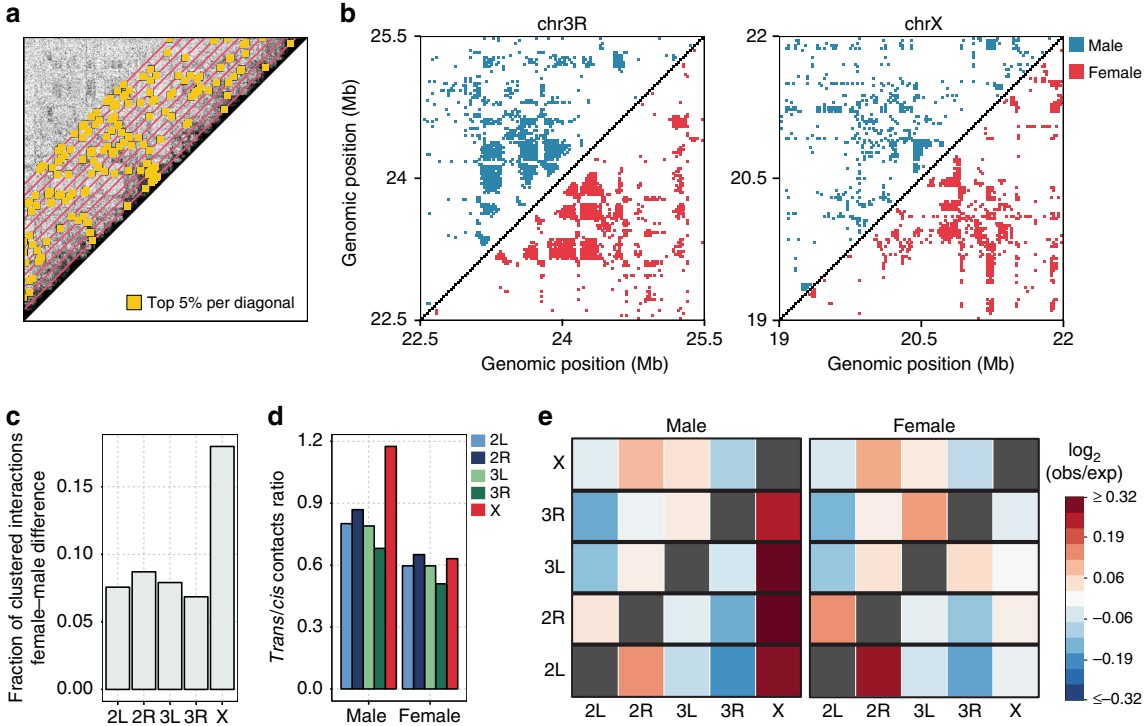

**Fig. 2** More dispersed *cis*-interactions and more *trans*-contacts in male chrX. **a** Cartoon to illustrate the selection of top-scoring interactions. In the normalized Hi-C contact map, for each diagonal (red lines), the data points with highest signal along the entire chromosome (top 5%, i.e. higher than 95th percentile, with default settings) are selected (yellow squares). **b** Representative (3 Mb) regions from contact matrices for an autosome arm (chr3R, left plot) and chrX (right plot). In each composite matrix, the top-scoring interactions (top 5%) are marked for male (blue points, upper triangle) and female (red points, lower triangle) sex-sorted Hi-C data at 25 kb bins resolution and normalized with chromosome-wise ICE. **c** The difference in the fraction of clustered top-scoring interactions (female–male) is shown for each autosome arm and chrX, based on 25 kb binned chromosome-wise normalized Hi-C data for sex-sorted embryos. Clustered top-scoring interactions were defined by aggregating neighbouring top-scoring data points (see Methods) and considering only distances up to 2 Mb. **d** The relative amount of *trans*-contacts for each chromosome is reported as the ratio of the number of *trans*- over *cis*-mapping read pairs for each chromosome in male and female sex-sorted embryos datasets. **e** The propensity of each chromosome to participate in *trans*-contacts is shown for both male (left) and female (right) samples. The *trans*-mapping read pairs originating from each chromosome (rows) are grouped based on the target chromosome (columns) and their number is compared (log$_2$ ratio) with the random expectation. In the random expectation model, *trans*-reads originating from any chromosome are expected to be uniformly distributed over the other (target) chromosomes, with their relative distribution depending on the size (and copy number) of each specific target chromosome (see Methods). The log$_2$ ratio between the observed and expected fraction of *trans*-read counts is reported as a colour gradient. Note that the heatmap is not expected to be symmetrical because the expected number of interactions is different depending on the respective size of the origin vs target chromosome in each pair. The diagonal is grey as *cis*-interactions are not considered.

(Supplementary Fig. 4c and f). Each of the chromatin state slopes is further shifted up in experimental data for male chrX compared with the female one (Supplementary Fig. 4c, e and f). These results suggest that the active chromatin state is itself prone to show more mid-/long-range interactions in experimental data. It is worth remarking that the dosage compensated chrX shows increased transcriptional activity as a consequence of the DCC action. Taken together these results indicate that the differential slope of interaction decay observed in male chrX (Fig. 1b) can be explained by a combination of lack of a paired homologous and more active chromatin.

All of these observations are compatible with the hypothesis that the unpaired male chrX is a more flexible and active chromatin fibre, prone to random fluctuations and interactions, even at long distances. In this regard, also the clustering of top-scoring interactions between female and male chromosomes is partially recapitulated in simulated data with or without homologous pairing (Supplementary Fig. 3f). However, the magnitude of difference in simulated data is smaller than in experimental data and is observed only at longer distances (≥5 Mb), thus attesting that pairing alone is not enough to explain the observed patterns. Local variations in chromatin accessibility

may also contribute to explain the differences in the interaction distribution.

**Local differences in structural domains**. We then asked if the dosage compensated chrX may also show differences in the structural domains. The literature in the field is generally concordant in reporting that TADs are mostly conserved across cell types[15,21], conditions[43] and even across species[44]. However, even subtle changes in genome compartmentalization at the sub-megabase scale can be relevant for cellular identity definition[45].

Thus, we analysed TADs on sex-sorted embryos Hi-C matrices, using multiple domain calling algorithms[15,46]. When comparing the domain boundaries identified in the male and female samples, a larger percentage of non-matching domain boundaries is observed specifically in chrX with respect to autosomes (Fig. 3a). A fraction of these differences are short range shifts in the called TAD borders (Supplementary Fig. 5a). This prompted us to achieve a more detailed definition of local changes in domains structure. Thus, we shifted the analysis to higher resolution matrices and applied a custom domain border calling procedure. As we needed a method not biased by copy number differences

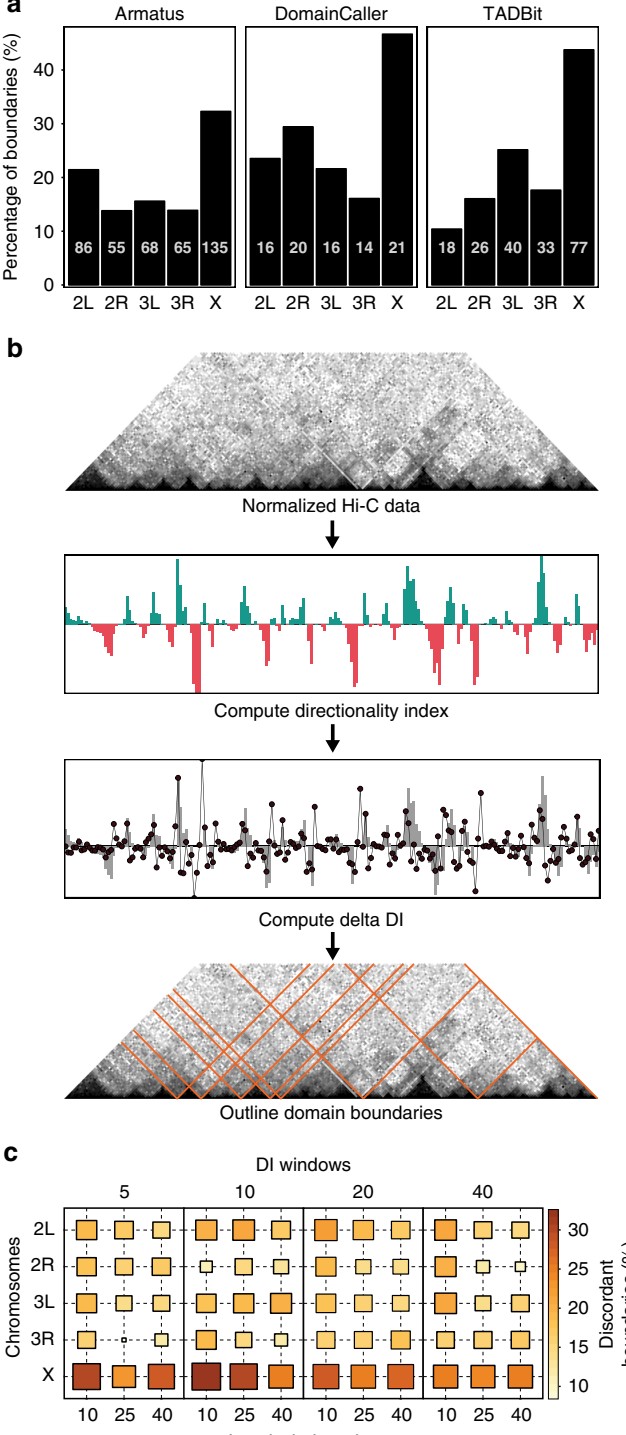

**Fig. 3** ChrX specific TAD borders differences between sexes. **a** The differences in TAD borders between sexes are reported for each chromosome (x-axis) as fraction (y-axis) of non-matching, i.e. not concordant, TAD borders when comparing male and female samples. TADs were called on 10 kb Hi-C matrices normalized with chromosome-wise ICE for male and female sex-sorted embryos using three alternative published algorithms: Armatus, DomainCaller and TADbit (as indicated). **b** Schematic overview of the Local Score Differentiator (LSD) algorithm for calling TAD borders. Starting from a normalized Hi-C contact matrix, we compute the directionality index (DI) over a window of size n. Next, the first derivative of DI (delta DI) is computed. Then a sliding window approach (local window of size m) is used to scan the delta DI vector to identify outliers within the local distribution as TAD boundaries. **c** Fraction of non-matching domain borders in male vs female samples based on LSD analysis of 10 kb bins Hi-C data normalized with chromosome-wise ICE. The proportion of non-matching boundaries is reported across variations of analysis parameters, including the size n of the directionality index window (upper axis) and the size m of the local window to scan for outliers (x-axis) across each chromosome (y-axis).

which is then used to find local extreme values to call domain borders (Fig. 3b) (see Methods).

We applied LSD to high-resolution (3.5 kb bins) Hi-C matrices of sex-sorted embryos and identified 851 boundaries on chrX (see Supplementary Data 2 for the complete list). ChrX consistently showed the highest proportion of non-matching domain boundaries independent of variations in analysis parameters (Fig. 3c), including alternative binning resolutions (5, 10, 25 kb) and normalization procedures (chromosome-wise ICE, genome-wide ICE and hicpipe-based explicit biases modelling) (Supplementary Fig. 5c). To confirm that the observed differences are not just due to higher noise as expected for the lower coverage on the single copy male chrX, we downsampled reads in the female Hi-C matrices to match the coverage in their male counterparts, chromosome by chromosome (Supplementary Fig. 5d).

**Dosage compensation mechanisms and insulators binding**. We then asked how the molecular mechanisms of DC are involved in the structural differences between male and female chrX. To this purpose, the differences between domain boundaries were qualitatively categorized as same (unchanged), appearing (boundaries absent in females but detected in dosage compensated males) or disappearing (detected in females but absent in males). Of the 851 boundaries detected in chrX, 377 (44.3%) boundaries were annotated as same, 174 (20.4%) as appearing and 300 (35.3%) as disappearing boundaries. Visual inspection of a representative region (Fig. 4a) shows that a number of disappearing domain borders seems to represent points of weakened insulation, with the female domain structure still partly visible in the male but with less sharp separation. We observed an enrichment of MSL binding sites near disappearing domain borders, which is consistent even if considering different definitions of MSL entry sites (defined as CES or HAS) from three different laboratories (Fig. 4b) (Supplementary Data 3). Furthermore, we also observed an enrichment of CLAMP, a binding partner of MSL, near disappearing domain boundaries (Fig. 4c) (Supplementary Data 4). We further investigated the distance relationship between MSL and CLAMP binding sites and the domain boundaries (Supplementary Fig. 6a). In total, 37.2% of disappearing, 28.6% of same and 25% of appearing boundaries harbour MSL or CLAMP binding sites within a distance of 7 kb (2 bins). We considered the two factors together as CLAMP sites were selected in previous literature[47] as co-located with MSL. In line with previous

and able to handle high-resolution data matrices, we developed a structural domain boundary calling procedure named Local Score Differentiator (LSD). LSD is robust to copy number differences, fast in analysing high-resolution matrices, and stably maintains good performances in terms of True Positive Rate (TPR) and False Discovery Rate (FDR) in benchmarking on simulated Hi-C data (Supplementary Fig. 5b). LSD builds upon the previously proposed Directionality Index (DI)[15] but segments the genome by defining thresholds on the local DI distribution, as opposed to the genome-wide DI distribution. Namely, delta DI vectors are computed to capture changes in the direction of the DI scores,

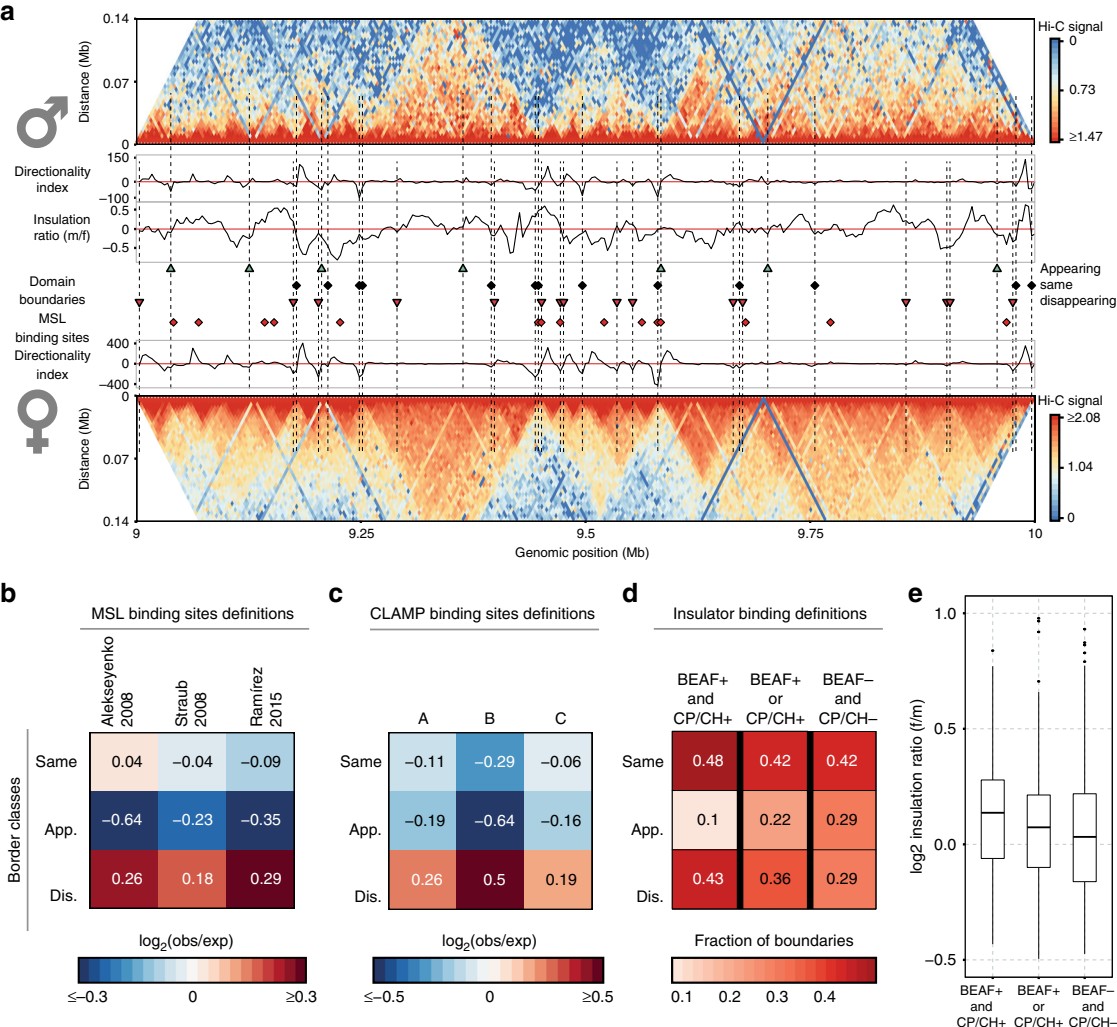

**Fig. 4** Changing domain borders are associated to dosage compensation features. **a** Example of domain borders identified by LSD classified by their concordance between male and female samples in sex-sorted embryos Hi-C data (binned at 3.5 kb, normalized by chromosome-wise ICE). A representative 1 Mb region of chrX is shown, with contact frequencies (colour bars on the side) for male (upper half) and female (lower half) data. The directionality index is shown adjacent to the Hi-C maps. Domain boundaries are marked based on their detection in male or female sample: disappearing (red triangles), appearing (green triangles) and same, i.e. unchanged (black diamonds) domain boundaries. MSL binding sites are also marked (red diamonds). The log_2 ratio of male over female insulation score is also shown in the middle track, above the domain boundary marks. **b** Enrichment (log_2 observed over expected ratio) of MSL binding sites around the domain borders grouped by differential pattern is shown. Three alternative definitions of MSL binding sites from three different laboratories were considered: Kuroda[4], Becker[5] and Akhtar[8] laboratories, respectively. The expected frequency was estimated assuming random uniform distribution of MSL binding sites along chrX. **c** Enrichment (log_2 observed over expected ratio) of CLAMP binding sites around the domain borders grouped by differential binding pattern is shown. Three groups of CLAMP binding sites as defined in Soruco et al.[47] are considered: MSL-dependent binding of CLAMP (A), Partially dependent on MSL binding (B) and MSL independent binding of CLAMP (C). The expected frequency was estimated assuming random uniform distribution of CLAMP binding sites along chrX. **d** After classification of boundaries on the basis of combinatorial presence of BEAF-32 and CP190 or Chromator (columns), for each class the proportion of boundaries grouped by differential pattern across sexes (rows) is shown. **e** The log_2 ratio of insulation scores (y axis, female over male ratio) is shown for each boundary class as defined by combinations of insulators. For each box the median is marked as horizontal line, the boxes mark the interquartile range (IQR), the whiskers extend up to 1.5 IQR and individual data points are shown for outliers beyond this range.

findings[8], we confirmed a network of pairwise interactions involving MSL binding sites in both male and female samples (Supplementary Fig. 6b). We also observed the same pattern for the CLAMP binding sites (Supplementary Fig. 6c).

As the disappearing boundaries showed an enrichment for MSL and CLAMP binding sites (Fig. 4b, c), we asked if one of the domain boundary classes shows a similar network of pairwise interactions. Instead, we found that the three boundary classes do not show a clear enrichment pattern for pairwise interactions (Supplementary Fig. 6d). However, when using only domain boundaries with a nearby (≤3.5 kb) MSL or CLAMP binding site, a very clear pairwise interaction pattern is observed in the disappearing boundaries class (Supplementary Fig. 6e). Whereas, a weaker enrichment pattern is observed in the boundaries class defined as same, and no clear enrichment observed in the appearing boundaries, although it must me noted that the latter is a less abundant group. This is in line with the preferential enrichment of MSL and CLAMP binding sites around disappearing boundaries, yet this result is explicitly showing that the association to the pairwise network of interaction is specifically

evident when disappearing boundaries are also coupled to MSL or CLAMP.

We then asked how the association between MSL or CLAMP to disappearing boundaries may also be related to chromatin insulators expected to be found at domain boundaries. Indeed, according to previous literature, chromatin insulators including BEAF-32, CP190 and Chromator are enriched at TAD boundaries in *D. melanogaster*[13]. Moreover, CP190 or Chromator, when found in conjunction with BEAF-32, can mediate long-range looping interactions[48]. Therefore, we investigated the MSL and CLAMP binding sites in relation to domain boundaries containing a CP190 or Chromator peak (CP/CH) in conjunction with a BEAF-32 peak (BEAF) (Supplementary Fig. 6f). Similar to before, disappearing boundaries containing a BEAF and CP/CH peak have more MSL and CLAMP binding sites nearby. In this case, the separation of the three domain boundaries classes is even more evident (Supplementary Fig. 6f).

Next, to investigate if differences in insulators binding may account for the male vs female differences in domain boundaries, we examined the chromatin marks landscape around changing domain borders[13] (Supplementary Fig. 7a). In most cases, domain boundaries harbour or are near to an insulator peak as detected by genome-wide chromatin immunoprecipitation (Supplementary Fig. 7b). Both the same and disappearing classes of domain borders show a similar pattern across male and female cells in terms of insulator peaks (Supplementary Fig. 7c), as well as in terms of genome-wide ChIP enrichment signal (Supplementary Fig. 7d). Instead appearing boundaries show lower density of insulator peaks (Supplementary Fig. 7c) and of genome-wide ChIP enrichment signal (Supplementary Fig. 7d).

These observations did not show a clear correlation between insulators binding and local differences in domain borders across sexes, although they hinted to a difference in the enrichment strength of insulators across the three domain border classes. Therefore, we reclassified domain boundaries based on the level of enrichment signal for BEAF-32, CP190 and Chromator (see Methods and Supplementary Fig. 8a). We grouped domain boundaries into three classes: (BEAF+ and CP/CH+) boundaries containing high signal for BEAF-32 together with CP190 or Chromator (CP/CH); (BEAF+ or CP/CH+) boundaries containing high signal for either one of BEAF-32 or CP/CH; and (BEAF− and CP/CH−) boundaries containing low signal for both BEAF-32 and CP/CH. The pairwise interaction profiles show a network of contacts specifically involving the (BEAF+ and CP/CH+) class of boundaries, similar to what previously observed for MSL and CLAMP binding sites (Supplementary Fig. 8b).

We then checked the overlap with the previous classification (Same, Appearing, Disappearing classes) and we found that the BEAF+ and CP/CH+ boundaries are mostly overlapping with boundaries which were classified as Same or Disappearing on chrX (Fig. 4d). This is specifically observed on chrX, as we verified that on autosomes the BEAF+ and CP/CH+ boundaries are prevalently overlapping with the conserved (Same) boundaries only (Supplementary Fig. 8c). Likewise, BEAF+ and CP/CH+ boundaries on chrX show a highly significant difference in insulation between males and females (Wilcoxon test p-value < 10^{-4}) (Fig. 4e and Supplementary Fig. 8e). This is not observed in autosomes, with the exception of some differences with lower statistical significance (Supplementary Fig. 8d, e). We also found that the BEAF+ and CP/CH+ boundaries are associated to MSL and CLAMP binding on chrX (Supplementary Fig. 8f, g).

Taken together, our observations indicate that three key insulators (BEAF-32, CP190, Chromator) are associated to a long-range interaction network on all chromosomes. On chrX, the insulation differences between sexes are specifically found at domain boundaries involved in this network. These chrX specific

differences in insulation are associated to MSL and CLAMP binding. This suggests an association between DC and differences in local chromatin compartmentalization.

**Domain insulation differences and chromatin accessibility.** To clarify the association between lower insulation at domain borders and DC, we examined chromatin accessibility in embryos and male and female cell lines, with or without knockdown of specific DCC components or co-factors.

First, we examined chromatin accessibility differences based on MNase-seq data after knockdown of *CLAMP* (in S2 and Kc cells) and *MSL2* (S2 cells only)[49]. Across all domain boundary classes, *CLAMP* knockdown specifically decreases accessibility (Wilcoxon test p-value < 10^{-4}) only on male chrX (Fig. 5a, b, Supplementary Fig. 9a). Instead, *MSL2* knockdown in male cell lines produces almost no change in autosomes nor chrX (Supplementary Fig. 9a).

Next, to confirm the association between lower insulation and higher accessibility we leveraged two independent 4C-seq based datasets[8,9], including 4C-seq profiles obtained with a total of 144 experiments across 28 probes (Supplementary Data 5). Considering data obtained from each 4C-seq probe, we examined chromatin interactions around the domain boundaries belonging to the three classes (BEAF+ and CP/CH+, BEAF+ or CP/CH+, BEAF− and CP/CH−) by generating metaprofiles for each probe and class (Supplementary Fig. 9b, c). BEAF+ and CP/CH+ boundaries tend to have higher 4C signal enrichment in the region proximal to domain boundaries, whereas BEAF− and CP/CH− boundaries have the lowest mean signal and BEAF+ or CP/CH+ boundaries have intermediate enrichment levels, both in male (S2) and female (Kc) cell lines. This is a general trend confirmed for all of the 4C probes (Fig. 5c). While BEAF+ and CP/CH+ boundaries have generally high 4C signal enrichment, it is even higher in male (S2) compared with female (Kc) samples (Wilcoxon test p-value < 10^{-4}) (Fig. 5c, right plot). These observations are consistent with a scenario where higher accessibility around BEAF+ and CP/CH+ borders results in lower insulation.

Although not statistically significant, similar patterns were observed in the second set of 4C probes, upon induction or inhibition of DC in female (Kc) and male (S2) cells, respectively[9]. This is achieved by RNAi silencing of *Sxl* in Kc cells (Kc^{ΔSXL}) or RNAi silencing of *MSL2* in S2 cell (S2^{ΔMSL}), respectively, compared with control samples (Kc^{GFP} or S2^{GFP}). Namely, we observed that 4C signal enrichment near BEAF+ and CP/CH+ boundaries is generally higher with dosage compensation (S2^{GFP} or Kc^{ΔSXL}) when compared with the non-dosage compensated condition (Kc^{GFP} or S2^{ΔMSL}) (Supplementary Fig. 9d).

We also noticed that BEAF+ and CP/CH+ boundaries are located near active genes (Fig. 5d). Therefore, we investigated in detail the changes in accessibility patterns at active gene transcription start sites (TSSs) (Supplementary Data 6) located near the boundaries (<10 kb). We leveraged single-cell ATAC-seq data from 12 h embryos[50] and DNase-seq data for cell lines by modENCODE[51]. We found that chrX gene TSSs located near BEAF+ and CP/CH+ boundaries show a statistically significant different accessibility between male and female in scATAC-seq data summarized at population level (Fig. 5e, Wilcoxon test p-value < 0.01) (see Methods). Likewise, DNase-seq data confirm highly significant differences in accessibility at BEAF+ and CP/CH+ boundaries (Fig. 5f Wilcoxon test p-value < 10^{-3}).

Then, we further considered BEAF+ and CP/CH+ chrX boundaries close to active gene TSSs and defined them as a core-set of domain boundaries (Supplementary Data 7). The 4C enrichment profile near the core-set boundaries is higher in male than in female

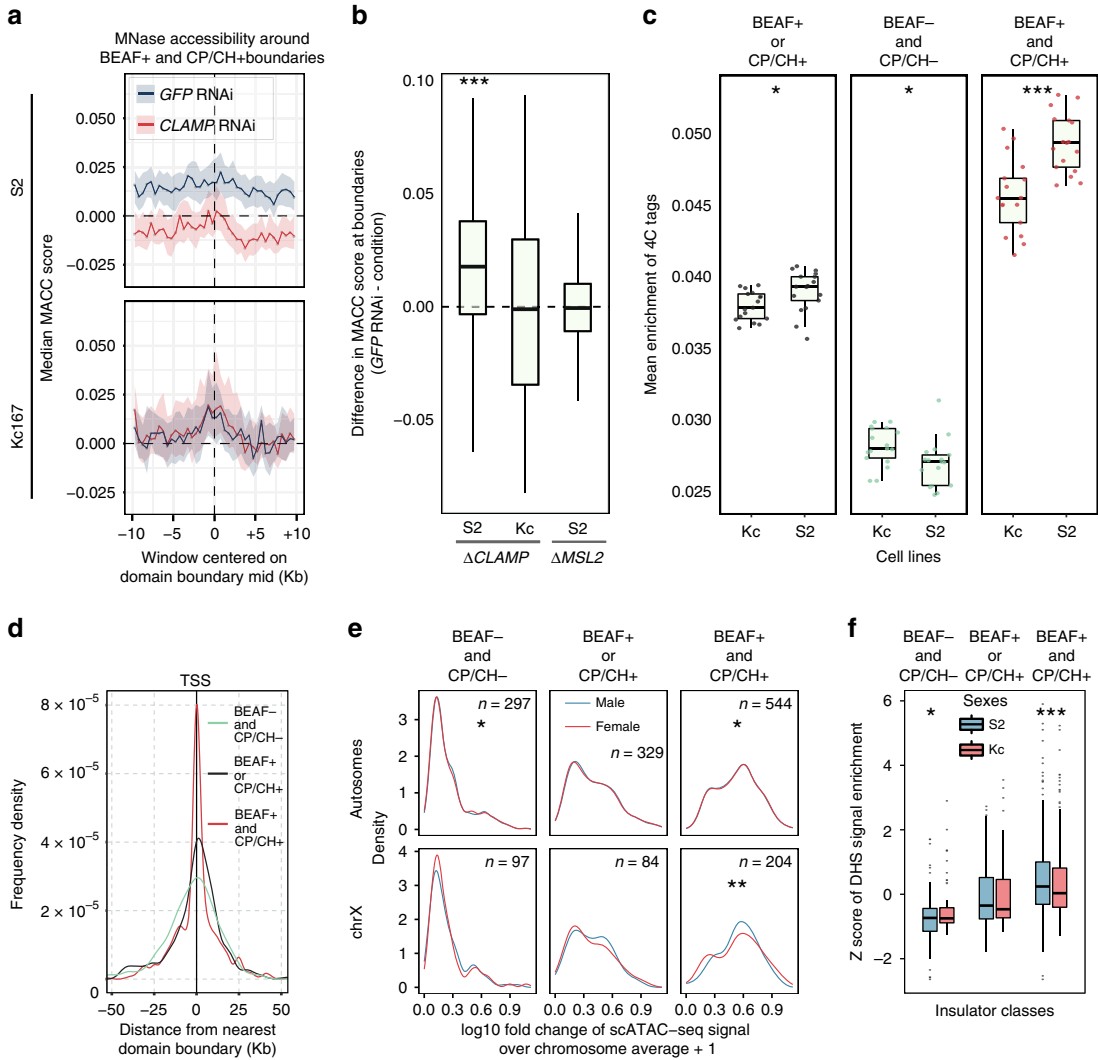

**Fig. 5** Independent datasets show higher accessibility of selected boundaries. **a** MNase-seq metaprofiles around BEAF+ and CP/CH+ boundaries based on data by Urban et al.[49]. Metaprofiles for both S2 (male, top) and Kc167 (female, bottom) cells are shown and labelled based on the treatment: *GFP* RNAi (black) and *CLAMP* RNAi (red) (40th to 60th percentile interval is shown as a shaded area). **b** The difference between *GFP* RNAi and *CLAMP* or *MSL2* RNAi MACC score at boundaries (y-axis) for each S2 or Kc cells sample (x-axis) (***Wilcoxon test *p*-value < 0.0001). For each box the median is marked as horizontal line, the boxes mark the interquartile range (IQR), the whiskers extend up to 1.5 IQR, individual data points for outliers beyond this range are omitted. **c** Mean 4C-seq enrichment signal around (±7 kb) each class of domain boundaries is based on metaprofiles for each 4C probe (each data point in the plot) from Ramirez et al.[8] in male (S2) and female (Kc) cell lines. The difference between S2 and Kc cells is reported (Wilcoxon *p*-value: ***<0.0001; *< 0.05). Boxplot elements are defined as for panel (**a**), all individual data points are shown. **d** For each domain boundary class (coloured lines) the distribution of distances to the nearest chrX active gene transcription start site (TSS) is shown. **e** scATAC-seq signal values in male and female embryos (fold change over chromosome average—see Methods) at the TSS of active genes grouped by the class of the nearest boundary (max ± 10 kb distance) is shown for autosomes (top) and chrX (bottom) for each boundary class. The difference between male and female is marked (Wilcoxon *p*-value: **<0.01; *<0.05). **f** DNase-seq signal (y-axis) (z-score transformed—see Methods) in cell lines (S2, Kc167) at the TSS of active genes grouped by class of the nearest boundary (max ±10 kb distance) is shown for each boundary class (x axis top) (Wilcoxon *p*-value: ***<0.01; *<0.05). Boxplot elements are defined as for panel (**a**), individual data points are shown for outliers beyond the whiskers range.

cell lines (Fig. 6a). Moreover, around the core-set boundaries (<10 kb) CLAMP shows a significantly higher binding signal in male than female cell lines (Wilcoxon test *p*-value < $10^{-4}$) (Fig. 6b).

Furthermore, we noticed that the average pairwise interaction profile for male embryos is noticeably higher than female embryos near the centre (±2 bins, i.e. 7 kb) (Fig. 6c left), as confirmed by the pairwise interaction profiles differences Fig. 6c right). This pattern is not observed for BEAF + and CP/CH + boundaries near active gene TSSs in autosomes (Supplementary Fig. 10) and indeed chrX shows significantly more positive differences than autosomes (Fig. 6d). This suggests that the core-

set boundaries tend to contact each other more in the dosage compensated male chrX than in the female embryos.

Finally, the male chrX specifically shows reduced insulation at the core-set of domain boundaries when compared with the female (Fig. 6e bottom), a pattern not observed in autosomes (Fig. 6e top).

Overall the presented results show male-specific chrX lower insulation around insulator bound chromatin regions participating in long-range interactions. This interaction network is stronger in males than in females. Herein, we show that the male chrX is generally more accessible at these sites, active genes

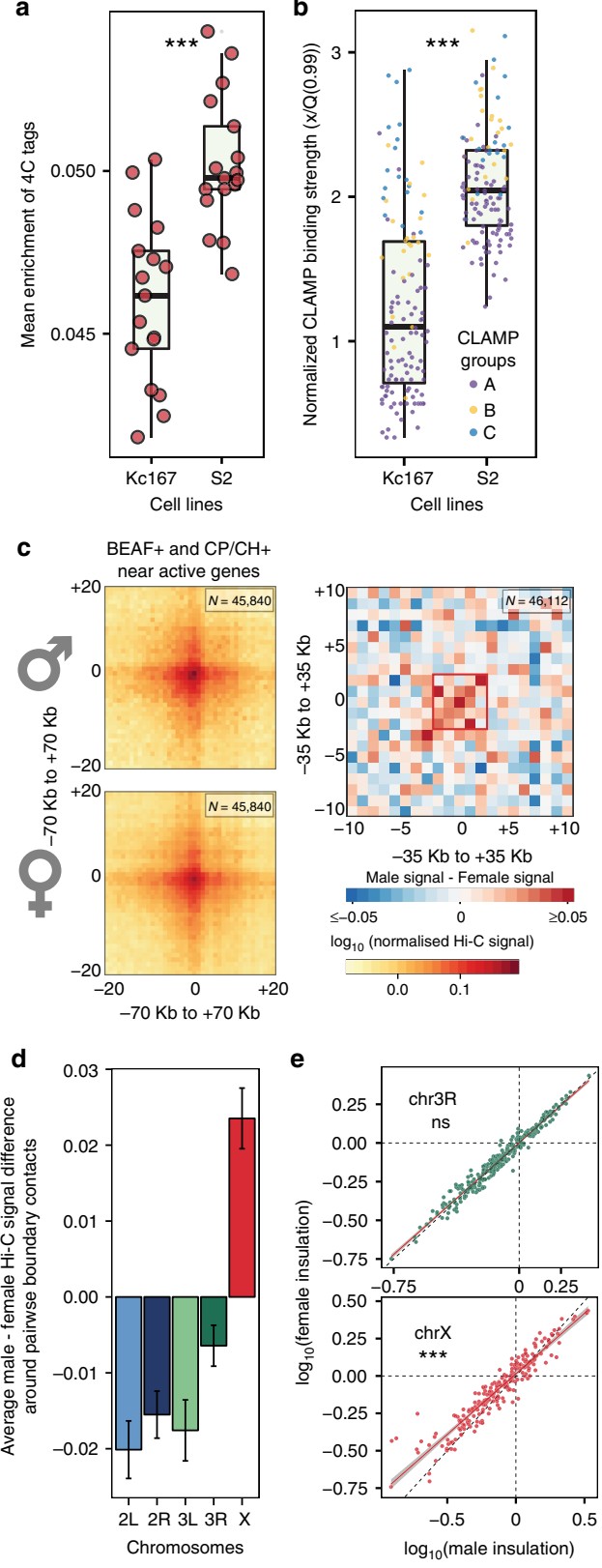

**Fig. 6** MSL associated CLAMP near BEAF+ and CP/CH+ boundaries. **a** Considering domain boundaries near active genes, 4C-seq metaprofiles were created for BEAF + and CP/CH + boundaries for each probe obtained from Ramirez et al.[8]. The mean 4C-seq enrichment around domain boundaries (±7 kb) is reported (y axis) for both S2 and Kc cells (***Wilcoxon p-value < 0.0001). **b** For each BEAF + and CP/CH + domain boundary near active genes, the nearest (max distance ≤ 10 kb) MSL associated CLAMP binding site was considered and its normalized maximum enrichment peak reported in the boxplot for S2 and Kc cells. The CLAMP binding strength in S2 is significantly higher (*** Wilcoxon p-value ≤ 0.001). For each box in panels (**a**) and (**b**) the median is marked as horizontal line, the boxes mark the interquartile range (IQR), the whiskers extend up to 1.5 IQR, all individual data points are shown. **c** Considering BEAF+ and CP/CH+ boundaries near active genes, average pairwise interaction profiles were created in 3.5 kb binned male and female Hi-C maps using a window of 70 kb (20 bins in 3.5 kb Hi-C maps) around boundaries (left). The male chrX average pairwise interaction profile showed noticeably higher signal. The average pairwise interaction profiles were recomputed using a window of 35 kb (10 bins in 3.5 kb Hi-C maps) and the difference (male–female) between the two average interaction profiles is shown (right). The male interaction profile shows higher interaction near the centre. **d** Using a ±2 bins window around the centre (similar to panel (**c**)), the average differences in the pairwise interactions between domain boudaries are shown for each autosome arm and chrX. Error bars are mean±standard error of the mean (SEM). For each chromosome, n is as follows: 2L = 28458; 2R = 41080; 3L = 30904; 3R = 68970; X = 46112. **e** Considering BEAF+ and CP/CH+ boundaries near active genes, the log_{10} insulation scores for each boundary is shown for chrX (bottom) and chr3R (top) for both male (x-axis) and female (y-axis) embryos. The observed linear fit (red) and expected (y = x, black dashed line) relationship between male and female insulation scores is also shown. The difference between insulation scores is highly significant for chrX (Wilcoxon test, p-value < 0.0001).

## Discussion

DC of chrX in *D. melanogaster* is a unique case of coordinated transcriptional upregulation of an entire chromosome. This epigenetically regulated mechanism is different from other widely studied DC models where the partial or complete silencing of chrX is used to balance the transcriptional output between sexes, as it happens in mammals and *C. elegans*. The inactivation of a single chrX in mammalian female cells is achieved with extensive rearrangement of chromatin, which results in remarkably evident alterations of chromatin architecture, as detected with Hi-C[14]. Similarly, the coordinated partial silencing of both copies of chrX in hermaphrodite *C. elegans* is detected as a change in the structural domain compartmentalization in Hi-C data[28]. Unlike these mechanisms of coordinated repression of transcription, in *D. melanogaster* DC is achieved by a coordinated upregulation of transcription. A recent article showed how the transcriptional activation in the *D. melanogaster* zygote genome results in the rapid definition of structural domains[52]. Yet the zygote genome activation is different from chrX DC, keeping inactive genes silent and specifically upregulating transcription only at active gene loci on chrX.

Previous literature suggested that global changes of chrX organization are associated to DC in fly[29]. Two recent articles using Hi-C data have highlighted the role of chromatin 3D organization in the MSL targeting of chrX and its subsequent spreading in *cis* to reach all of the active genes[8,9]. However, neither study could identify large-scale differences in chrX architecture in male and female flies, despite the expectations based on previous literature.

are preferentially found near these regions and the accessibility near the TSSs of such genes is higher in males than in females. Our observations support a model where local changes in chromatin accessibility associated to DC are reflected in the Hi-C contact profile with lower insulation around specific domain boundaries.

It is worth remarking that chrX DC in fly poses exceptional challenges for standard methods of Hi-C data analyses, as they are generally not designed to account for the specific confounding factors of *Drosophila* DC. First of all, the male chrX is present in single copy, thus on average it presents half as much signal as the female chrX. This requires the adoption of targeted chromosome-wise procedures for normalization and data analysis. Also, the transcriptional upregulation is not uniformly affecting chrX as the transcriptional status of silent genes remains unchanged. Therefore, local changes in chromosomes structure must be investigated further.

We were able to detect specific differences in the global interaction profile of chrX in male and female flies thanks to the development of ad hoc data analysis solutions. In particular the non-parametric selection of top-scoring interactions and the polymer folding simulations were instrumental in confirming the reported observations. We observed a small yet robust, reproducible and significant difference in the Hi-C interactions decay as a function of genomic distance. Namely, the dosage compensated male chrX engages in more mid-/long-range interactions that occur in a non-clustered pattern. This suggests that these interactions result from random contacts rather than stable functional interactions. It may be worth remarking that this result is not in contrast with the stronger pairwise interaction between the core-set domain boundaries shown in Fig. 6, as those are just a selected subset of specific interactions found in both male and female chrX, yet a bit stronger in the male. Polymer folding simulations confirm that the lack of a homologous paired chromosome may partially explain the increased mid-/long-range contacts, but can only partially recapitulate the pattern of scattered top-scoring interactions. All of our observations on interaction profile differences indicate that the unpaired and dosage compensated chrX is a more flexible chromatin fibre, prone to random contacts even at mid- and long-range distances.

We then focused on differences in chrX structural domains between sexes. This problem required as well the development of custom solutions to overcome the limitations and copy number-related biases of existing methods to identify TADs. When our study began, most TAD calling procedures either used chromosome-wise or genome-wide defined thresholds. We instead devised a structural domain border calling procedure (LSD) that is sensitive to local changes in the chromatin fibre. LSD is accurate and fast, thus allowing us to call boundaries in high-resolution matrices and to identify domain borders differences between sexes. In particular, we identified a set of weakened (Disappearing) domain borders in male compared with female flies. These domain borders are associated to strong binding of insulators known to orchestrate long-range contacts[48], namely BEAF-32 in conjunction with CP190 or Chromator. This network of long-range contacts is conserved across sexes, however, in male chrX there is a detectable increase in pairwise contacts around these sites, when restricting to active gene TSSs. These sites are also associated to binding of DCC and its co-factor CLAMP, they co-localize with active genes, have reduced insulation and their surrounding regions show a general increase in accessibility in the male compared with the female. These observations are confirmed across several data types including Hi-C, 4C, ATAC-seq, DNase-seq, MNase-seq, including data on dynamic establishment (in female cells) or inhibition (in male cells) of DC as well as models of CLAMP silencing. Thus, our analyses confirm an association between changes in chromatin accessibility and weakening boundaries. Our findings complement recent literature reporting that MSL associated CLAMP binding can induce a local increase in chromatin accessibility. This, together with the strong binding of insulators (BEAF−32, CP190, Chromator) could explain the structural preservation of topological domains between male and female chrX, yet with localized changes in insulation reflected in the Hi-C contact matrices.

In conclusion, this work shows that DC of fly chrX results in global and local conformation differences as hypothesized based on previous literature. We demonstrated that these differences in 3D chromatin architecture are subtle but detectable with chromosome conformation capture techniques. This is the first report of these chromatin architecture differences on a genome-wide scale, as most recent literature adopting Hi-C to characterize targeting and spreading of DCC in *D. melanogaster* failed to detect such global structural changes. We achieve these results thanks to the development of custom methodological solutions for data analysis, specifically tailored to the peculiar characteristics of the biological problem under investigation. Overall, we remark the importance of coupling carefully designed Hi-C data analysis procedures to specific biological questions.

## Methods

**Hi-C data processing**. Hi-C data were processed with the hiclib (2016-07-14 version—commit fe3817a; https://bitbucket.org/mirnylab/hiclib) and cooler (v0.3.0; https://github.com/mirnylab/cooler) packages by Leonid Mirny's lab for ICE normalization[31]. When explicitly indicated, as alternative for specific analyses hicpipe (v1.03) was used for the probabilistic bias modelling normalization proposed by Yaffe and Tanay[32].

For both pipelines (ICE and hicpipe) the sequencing reads were aligned to the dm3 genome build considering only chromosomes X, 2 and 3. Chromosome 4, Y and the heterochromatic portions (named with suffix Het) were left out. The rationale behind this solution is that chr4, chrY and all of the Het portions of other chromosomes are very short in size, thus not comparable to chrX and other autosome arms when assessing metrics like interaction decay as distance. Moreover, they are known to be characterized by a strongly heterochromatic epigenetic status[51,53]. Thus, they may not be representative of the average autosomal genomic regions to be used as reference comparison for dosage compensated chrX (see also Johansson el al.[54]).

hicpipe used bowtie[55] (v1.1.2) for alignment. hicpipe was used with default parameters, except SEGMENT_LEN_THRESHOLD was set to 800 for the sex-sorted embryos dataset. We set this parameter after examining the distribution of the sum of distances between read pairs and their nearest downstream fragment end.

For ICE pipeline we used the iterative_mapping module in hiclib for aligning reads to the reference genome. hiclib alignments were run with bowtie2[56] version 2.2.9. For the sex-sorted embryos data (GSE94115) we adopted the following parameters: min_seq_len = 20, len_step = 10, seq_start = 0 and seq_end = 49. For S2 and clone-8 cell lines data obtained from Ramirez et al.[8] we used: min_seq_len = 20, len_step = 10, seq_start = 0 and seq_end = 50. For Kc167 cell line data from Li et al.[30]: min_seq_len = 20, len_step = 10, seq_start = 0 and seq_end = 50. Additional bowtie2 flags were --mm and --very-sensitive.

The following filtering parameters were applied for hiclib: For embryos, S2 and clone-8 samples the maximum molecule length was set to 800, for Kc167 samples maximum molecule length was set to 300 (as in the original publication). We also filtered for duplicates using the filterDuplicates function. Afterwards all technical replicates were merged into their corresponding sample. In most of the analyses, the biological replicates of male or female sex-sorted embryos were merged, as we verified their reproducibility by computing the stratum-adjusted correlation coefficient (SCC) with the hicrep tool[57]. The sex-sorted embryos replicates show an SCC value of 0.97 for the male and 0.98 for the female samples. The final read numbers are available in Supplementary Table 1.

We summarized the Hi-C data at several resolutions (bin sizes), including 25, 10 and 3.5 kb. At the highest resolution (3.5-kb bins) we verified that in the Hi-C maps at least 80% of the bins had at least 1000 reads as proposed by Rao et al.[20]. Finally, the binned matrices were subjected to chromosome-wise ICE normalization using mirnylib.numutils.iterativeCorrection and genome-wide ICE normalization using cooler iterative_correction. To accept rows or columns for normalization we required at least 40 as sum of read counts. Moreover, to remove non-informative read pairs we excluded the first two diagonals during normalization (interactions at distances 0 or 1 bin). Lastly, the tolerance value set to 1e−02.

**Computing decay of Hi-C signal**. We considered all interactions at distances ranging from 2 bins (50 kb for 25 kb matrices) to 100 bins (2.5 Mb for 25 kb matrices). In normalized Hi-C matrices, NAs, NaNs and infinite values were set to 0. The median Hi-C signal (y-axis) was computed at each distance (x-axis).

When indicated, the Hi-C signal was transformed into contact probabilities (contact frequencies) by assuming the contact probability is maximum (=1) when considering neighbouring genomic loci. To this concern the median normalized Hi-C signal is computed for each diagonal and divided by the median signal at the first informative diagonal (2 bins distance) to obtain contact frequencies. Then the

median contact frequencies are $\log_{10}$ transformed ($y$-axis) to be plot against the log of genomic distance ($x$-axis) in the log–log plots. This procedure is applied in Fig. 1b and Supplementary Fig. 1a and c. The original median $\log_{10}$ Hi-C normalized signal values relating to Fig. 1b can be observed in Supplementary Fig. 2a genome-wide ICE.

Previous literature proposed an alternative probabilistic transformation of Hi-C matrices[33], based on the same assumptions of maximum contact probability near the diagonal. We also applied this transformation where the signal inside every cell of the Hi-C matrix is divided by the mean normalized signal at the first informative diagonal (2 bins distance) to obtain a contact probability. Any resulting value >1 was set to 1. This method is only applied to Supplementary Fig. 2b.

We then used the lm function in R to fit a linear model to the values in the log–log plot to obtain the slope coefficient. The linear model fitting was done for values at distances ranging from 2 bins (50 kb or 4.69 in the $\log_{10}$ scale) to 15 bins (375 kb or 5.57 in the log scale), i.e. in a range of distances where the decay is close to linear in the log–log plot.

The interaction decay differences are assessed by computing the pairwise differences of slope coefficients (deltas) between autosomes or between chrX and autosomomes. The slope coefficient deltas of chrX vs autosomes are then compared with those between autosome pairs using Wilcoxon test as indicated in individual boxplots.

Alternatively, to assess the difference between the interaction frequency plots, the cumulative density functions (CDFs) of the interaction probability for autosomes or chrX are computed. CDFs of interaction probability were estimated from 50 kb to 2.5 Mb as cumulative sums of median Hi-C contact frequencies for each distance, then divided by the cumulative sum maximum value to make it equal to probability 1. Kuiper's statistic for pairwise comparisons between autosomes or between chrX and autosomomes is then computed as the sum of absolute values for the maximum positive and negative differences between CDFs for the two compared chromosomes as

$$V = \max(\text{cdf}_{\text{chr}a} - \text{cdf}_{\text{chr}b}) + \max(\text{cdf}_{\text{chr}b} - \text{cdf}_{\text{chr}a}) \quad (1)$$

where chr$a$ and chr$b$ indicate the pair of chromosomes $a$ and $b$ considered in each comparison. The difference in the estimated pairwise Kuiper's statistics of chrX vs autosomes are then compared with those between autosome pairs using Wilcoxon test as indicated in individual figure panels.

**Downsampling of Hi-C matrices**. To account for the disparities in coverage due to copy number and sequencing depth differences between male and female samples, we used two approaches for downsampling of read counts, as indicated in the results. In the first, we simulated the effect of single copy number on male autosomes by randomly down sampling 50% of the autosomal reads in the male samples. For this we used the numpy.random.binomial function in python with the probability parameter set to 0.5. The downsampled observed read counts were then normalized using chromosome-wise ICE. This method was used to check the rate of Hi-C decay when the copy number of autosomes is similar to that of chrX (Supplementary Fig. 1b).

In the second approach, we downsampled all female chromosomes (observed *cis*-read counts) by the ratio of *cis*-interactions count present in the corresponding male chromosome (Supplementary Data 1), to make the total sum of observed *cis*-read counts comparable between male and female samples, chromosome by chromosome. The downsampled observed read counts were normalized with chromosome-wise ICE. This approach was used to verify the effect on TAD calls, and the effect on clustering of top-scoring interactions between the male and downsampled female samples (Supplementary Fig. 5d and Supplementary Fig. 3a, respectively).

**Polymer folding simulation**. We modelled four chromosomes (two chromosomes 3R and two chromosomes X) by semi-flexible self-avoiding block copolymers as described in Ghosh and Jost[40]. Each chromosome consisted of $N = 3200$ beads of 10 kb, each of size $b$. The four polymers moved within a cubic box of size $L_x \times L_y \times L_z$ under periodic boundary conditions in the $x$–$y$ plane and rigid wall condition in the $z$ direction to mimic the nuclear membrane.

To each 10 kb bead $i$, we assigned an epigenetic state $e_i$ based on the classification given by Filion et al.[57] (active, PcG, inactive and heterochromatin) where, for simplification, we merged the two active states into one unique class (see below). For chromosome 3R, to the 2790 beads that correspond to mappable regions, we added 410 heterochromatin beads to model parts of the centromeric and pericentromeric regions. Similarly, for chromosome X, 958 heterochromatin beads were inserted at the end of the 2242-long chain. The energy of a given configuration was then given by:

$$H = \sum_{\text{chr}=1}^{4} \frac{k}{2} \sum_{i=1}^{N-1} \left(1 - \cos\theta_{i_{\text{chr}}}\right) + \varepsilon \sum_{\text{chr,chr}',i,j} \delta_{i_{\text{chr}} j_{\text{chr}'}} \Delta_{e_{i_{\text{chr}}},e_{j_{\text{chr}'}}} \quad (2)$$

With $k$ the bending rigidity, $\theta_i$ the angle between the bond vectors $i_{\text{chr}}$ and $i_{\text{chr}}+1$ of chromosome chr, $\varepsilon$ (<0) the contact energy between beads of the same epigenetic state, $\delta_{i_{\text{chr}} j_{\text{chr}'}} = 1$ if beads $i$ from chromosome chr and $j$ from chr$'$ occupy nearest-neighbour sites (= 0 otherwise) and $\Delta_{e_{i_{\text{chr}}},e_{j_{\text{chr}'}}} = 1$ if their epigenetic state is equal (= 0 otherwise).

Homologous pairing (between the two copies of chromosome 3R and, in the female-like situation, between the two copies of chromosome X) was accounted for by first initially forcing the homologous polymers to occupy the same position in the simulation box and then, in the rest of the simulation, by forcing some homologous beads to stay in contact. We randomly selected these forced beads so that the proportion of homologous loci in close contact, for every epigenetic classes, is the one observed by Abed et al.[38] using Hi-C (~100% for active, ~75% for PcG, ~50% for inactive and ~50% for heterochromatin). In the male-like situation (unpaired Xs), the two X copies started from different locations and no forcing was imposed. In this case, a global pairing, putatively driven by epigenetic interactions, was never observed.

The dynamics of the chains followed a simple kinetic Monte–Carlo scheme with local moves using a Metropolis criterion applied to $H$. The values of $k$ (= 1.5 kT), $b$ (= 105 nm), $\varepsilon$ (= −0.1 kT), $L_x = L_y$ (= 2 μm) and $L_z$ (= 4 μm) were fixed using the coarse-graining and time-mapping strategies developed in Ghosh and Jost[40] for a 10-nm fibre model and a volumic density = 0.009 bp/nm$^3$ typical of *Drosophila* nuclei. In both situations (female-like and male-like), 300 independent trajectories were simulated starting from random, compact, knot-free initial configurations with chromosomes aligned in a Rabl-like manner (all centromeric regions at the bottom of the simulation box). Each trajectory represented ~13 h of real time. Note that our simulations is able to catch the formation of chromosome territories, compartments and TADs as well as homologous pairing. Predicted Hi-C data were generated by sampling 10 configurations per trajectory every 20 min after 10 h (the typical time after the last mitosis for late embryos), two monomers being in 'contact' if their relative distance was less than 160 nm.

**Defining chromatin state domains for interaction decay plots**. Epigenetic domains (chromatin states) information was obtained from Filion et al.[58]. Five domain colours were defined in the original publication: BLACK, BLUE, RED, YELLOW and GREEN. BLACK corresponds to inactive chromatin regions, BLUE corresponds to Polycomb group proteins (PcG) regulated domains, RED and YELLOW correspond to actively transcribed regions, and GREEN corresponds to centromeric or pericentromeric heterochromatic regions. In this study, the GREEN regions were ignored, the RED and YELLOW regions were merged into one single group named RED regions representing actively transcribed domains. Furthermore, all the individual domains segment smaller than the binning resolution of the Hi-C data were removed. For the experimental Hi-C data this value is 25 kb, whereas for the polymer folding model-based simulated contact matrices this value is 20 kb (see results reported in Supplementary Fig. 4). Finally, interaction decays were computed as reported above.

**Selection and clustering of top-scoring interactions**. For the non-parametric selection of top-scoring interactions we used normalized Hi-C data binned at 25 kb bins. NAs, NaNs and Infinite values were set to 0 and we discarded the first two diagonals (interactions occurring at distances 0 or 1 bin). We then selected the highest 5% (default threshold, applied unless otherwise specified) of normalized Hi-C contact values in any given diagonal as the top-scoring interactions (Fig. 2a). When indicated, different thresholds were adopted as percentage of highest scoring interactions, as well as thresholds on the maximum distance of interacting loci pairs.

To define clustered top-scoring interactions we consider the euclidean distance between any pair of top-scoring interactions ($i$, $j$) with coordinates ($i_x$, $i_y$) and ($j_x$, $j_y$), respectively, in the space of Hi-C matrix bins coordinates. With bin size 25 kb, the distance $D$ for each pair is defined as:

$$D = \left(\sqrt[2]{(i_x - j_x)^2 + (i_y - j_y)^2}\right) \times 25,000 \quad (3)$$

If distance $D \leq 25$ kb interaction $j$ and $i$ are grouped under the same cluster name. During merging, in an iterative process the list of clusters is scanned and clusters sharing elements are merged into larger clusters. Finally, we obtain a list of clusters containing unique interactions. We report the difference in the proportion of clustered top-scoring points. With default settings ($D \leq 25$ kb) the procedure is equivalent to cluster neighbouring top-scoring interactions only.

**Propensity of chromosomes to participate in *trans*-contacts**. For each chromosome pair (a,b), where $a \neq b$ are chromosomes {$2L, 2R, 3L, 3R, X$} the expected number of *trans*-contacts is estimated with a null model where *trans*-contacts originating from any chromosome are uniformly distributed over the other chromosomes (targets). This is estimated by adjusting the expected counts by the target chromosomes length and copy number. For example, the expected *trans*-contacts $E_{2L,2R}$ originating from chr2L and targeting chr2R is estimated as:

$$E_{2L,2R} = \frac{(c_{2R} \times l_{2R})}{\sum (c_b \times l_b)} \times T_{2L} \quad (4)$$

where $b$ contains the set of target chromosomes {$2R, 3L, 3R, X$} (all except the origin chromosome 2L). Whereas $c_i$ and $l_i$ are the expected copy number and length, respectively, of the specified chromosome $i$. Then $T_{2L}$ is the total number of *trans*-contacts originating from the chromosome 2L.

**Defining domain boundaries in 3.5-kb bins using LSD**. Domain boundaries have been defined on 3.5 kb bins matrices using Local Score Differentiator (LSD) (code available at https://bitbucket.org/koustavpal1988/fly_dc_structuralchanges_2018/). The directionality index (DI values) was computed as in Dixon et al.[15] on a window of 35 kb (10 bins) using the ComputeDirectionalityIndex function. We then computed the forward and backward differences of the DIs using the Forwards. Difference and Backwards.Difference functions defined as the difference in DIs between a bin and its adjacent downstream or upstream bin, respectively.

$$\Delta DI_{forward} = DI_i - DI_{i+1} \tag{5}$$

$$\Delta DI_{backward} = DI_i - DI_{i-1} \tag{6}$$

We then identify domain starts and domain ends using the outliers of the forward and backward differences within a local window of 25 bins (upstream and downstream) corresponding to 175 kb in a 3.5 kb binned matrix. Outliers are detected as follows:

First, we define fences on the forward and backward differences distribution as

$$Fence_{forward} = Q(\Delta DI_{forward}, 0.25) - 1.5 \times (Q(\Delta DI_{forward}, 0.75) - Q(\Delta DI_{forward}, 0.25)) \tag{7}$$

$$Fence_{backward} = Q(\Delta DI_{backward}, 0.75) + 1.5 \times (Q(\Delta DI_{backward}, 0.75) - Q(\Delta DI_{backward}, 0.25)) \tag{8}$$

where, $Q(\Delta DI, 0.75) - Q(\Delta DI, 0.25)$ is the interquartile range $\Delta DI$, $Q(\Delta DI, 0.25)$ and $Q(\Delta DI, 0.75)$ are the 25th and 75th quantiles of the $\Delta DI$ distributions within the window. 1.5 is the Tukey's constant used to select outliers in the local window values distribution.

Domain starts require the DI value to be finite, $\Delta DI_{forward} \leq Fence_{forward}$ and $\Delta DI_{forward} \leq DI$. Domain ends require the DI value to be finite, $\Delta DI_{backward} \geq Fence_{backward}$ and $\Delta DI_{backward} \geq DI$.

An additional filter, requiring $DI \leq 0$ for domain starts, and $DI \geq 0$ for domain ends is also applied for a stricter definition of boundaries (strict parameter). This parameter was set to FALSE (strict = FALSE) in the analyses for this study, unless otherwise noted. LSD by default also attempts to fill in any gaps that may exist between two called domains by connecting the end and start of two consecutive domains (Fill.gaps parameter), this parameter was set to FALSE (Fill.gaps = FALSE) in the analyses for this study, unless otherwise noted.

As LSD identifies domain starts and ends separately, a list of unique domain end positions is considered and extended on both sides by half bin size to obtain bins spanning adjacent start and end bins as reference border region for downstream analyses. We used the MakeBoundaries function to carry out this transformation and obtain 3.5 kb (equal to bin size) wide domain border regions. For boundaries list see Supplementary Data 2.

**Defining domain boundaries using other TAD callers**. Armatus[59] (v2.1) TAD caller was obtained from https://github.com/kingsfordgroup/armatus, and run with the parameters −r specifying the resolution (10 kb), −g specifying gamma values ranging from 0.1 to 1 with 0.1 step {.1, .2, ..., 1} and −m.

DomainCaller[15] was obtained from the public repository by the original authors (http://bioinformatics-renlab.ucsd.edu/collaborations/sid/domaincall_software.zip) and was run with directionality index computed at 2 Mb distance on 10 kb matrices. As previously reported by multiple groups[20,36] the original code was affected by a problem causing the programme to exit due to a division by zero in random generated numbers that may occur randomly with larger matrices. To circumvent this problem we used the patch as proposed in Forcato et al.[36], where the programme reiterates the random number generation.

TADBit[46] (v0.1_alpha.360) (https://github.com/3DGenomes/TADbit) was executed using default parameters on uncorrected counts on 10 kb bins matrices.

In all three cases, we computed the proportion of non-matching domain boundaries using as reference the list of TAD starts produced by the TAD callers.

**Defining boundary change annotations**. We used exact match of domain boundaries, i.e. intersection of the lists of genomic bins marking the boundary, to classify boundaries as disappearing, appearing or unchanged between the male and female samples. For boundary change annotations see Supplementary Data 2.

**Analysis of pairwise Hi-C contacts between selected loci**. A common approach used to summarize the average pairwise Hi-C contacts between a set of selected genomic loci is the so-called PE-SCan plot[60], later applied also by Ramirez et al.[8] to summarize the pairwise contacts between HAS in D. melanogaster. We applied a similar approach for summarizing pairwise Hi-C contacts between selected regions in Fig. 6c and similar panels in Supplementary Figs. 6 and 8.

Briefly, starting from a set of loci R = {$r_1, r_2, r_3, ...$}, a window of size $n$ was created around pairwise combinations of $r_{ij}$, where $i$ and $j$ are numeric indices for genomic bins in a Hi-C matrix. To avoid overlap of windows between each other and overlap with the beginning or end of chromosomes, the $i$ and $j$ indices are

filtered to satisfy the following criteria:

$$1 + n \leq i \leq (D - n) \tag{9}$$

and

$$i + 2n < j \leq (D - n) \tag{10}$$

where $D$ corresponds to the dimension (maximum bin index) or size of a symmetric Hi-C matrix containing the intra-chromosomal contacts for any chromosome arm considered (2L, 2R, 3L, 3R or X). The window of size $n$ is created on the row and column indices, $i$, $j$ respectively. In our case, $n = 20$ and the binning resolution of the Hi-C matrix was 3.5 kb. Therefore, the examined window is of size ±70 or 140 kb surrounding any given pair of loci $r_{ij}$. The Hi-C matrix is subset using these windows to obtain a matrix $Mr_{ij}$ for each $r_{ij}$. For each $r_{ij}$ an extended matrix subset is also considered for local background normalization. In this case, a window of size $n + 1$ is used, to obtain $Mr_{ij}$. In both $Mr_{ij}$ and $Mr_{ij}$, NAs and Infinites were set to 0. The matrix $Mr_{ij}$ is then normalized (divided) by the mean interaction frequency of $Mr_{ij}$, to obtain the local background adjusted matrix $Mr_{ij}$, which is then transformed in log2 scale. For each pair $r_{ij}$ we aggregated all $Mr_{ij}$ by computing their mean background adjusted interaction frequency MR for the selected list of loci.

**Insulators binding at domain boundaries**. Insulators binding peaks obtained from ChIP-chip experiments were first queried on modMine and downloaded from the modENCODE data repository[61]. In particular, we used BEAF-32, CP190, Chromator and CTCF in Kc167 (respective IDs: 3745, 3748, 277, 908), in S2 (respective IDs: 274, 925, 279, 3281) and BEAF-32, CP190 and CTCF in embryos (5130, 5131, 5057, respectively).

The binding peaks were overlapped to the 3.5 kb binning table associated with the chromosome-wise ICE normalized Hi-C matrices using the GenomicRanges package[62]. The number of overlaps per bin was counted for each peak file using the countOverlaps function.

We then created a 10 bin (35 kb) window around the domain boundaries. To do so, we considered mid-point of the bins as reference coordinate. Boundaries at <35 kb distance from the start and end of the chromosome were removed. Then we aggregated the peaks count per bin for each insulator and boundary class, and the counts were averaged. Finally, for visualization we applied spline smoothing as implemented in the ggplot package (geom smooth, glm method with natural cubic spline and 10 degrees of freedom).

To compute the median insulators enrichment around domain boundaries, we used the same ChIP-chip datasets listed above, for which we retrieved the enrichment signal (.wig) files from the modENCODE data repository. Signal files (.wig) were rescaled by dividing the signals in each file by their 99th percentile, to facilitate comparisons across datasets accounting for potential differences in ChIP efficiency. Insulator average profiles were calculated using deepTools[63] (version 2.5.3). Each average profile is displayed in a 10 kb window centred on domain boundaries, with a bin size of 100 bp.

**Combinatorial presence of insulators at boundaries**. The strength of insulator binding associated to each domain boundary was defined based on the maximum ChIP signal intensity in the scaled .wig files (see previous heading) within the boundary genomic window. We call these values insulator summits. As the correlation of insulator summits was generally high between male and female, for each domain boundary we considered just the maximum between male and female insulator summits, in order to classify the domain boundaries based on the combinatorial presence of insulators.

For each insulator, the summit values had a bimodal distribution, thus we selected the mid-point between modes as a cut-off to discriminate high and low signal. This mid-point was around 0.5 in the scaled ChIP signal intensity for each insulator. We used this threshold to classify as high or low binding each domain boundary region.

Then we classified boundaries based on the combination of insulators as:

i. (BEAF+ and CP/CH+) class, with high signal for BEAF-32 and high signal of at least one of CP190 or Chromator;

ii. (BEAF+ or CP/CH+) class, with high signal for BEAF-32 and low signal of both CP190 and Chromator or the opposite, i.e. low signal of BEAF-32 and high signal of either CP190 or Chromator;

iii. (BEAF− and CP/CH−) class, with low signal of BEAF-32 and low signal of both CP190 and Chromator.

**Analysis of MNase-seq profiles around domain boundaries**. Processed MNase-seq data for S2 and Kc167 cell lines with knockdown of CLAMP, MSL2 and control GFP was obtained from Urban et al.[49] (GEO dataset ID: GSE99893). Two replicates were present for each sample and each file contains MNase accessibility score (MACC) signal values for the D. melanogaster genome binned at 100 bp. The signal was averaged across replicates to get the signal corresponding to each 100 bp bin for each sample.

A 20 kb window (±10 kb) was created around each boundary mid-point and the window was binned at 500 bp bins (or 40 genomic intervals). The mid-point of

each 100 bp bin in the MNase-seq data files was mapped to the 500 bp genomic intervals centred around a domain boundary, and the average MACC score for each 500 bp interval was computed.

Finally, for each boundary class, i.e. (BEAF+ and CP/CH+), (BEAF+ or CP/CH+), (BEAF− and CP/CH−) for each chromosome X or autosome arms (2L, 2R, 3L, 3R) all such windows were aggregated and a median MACC profile was computed for each sample. In order to take into account the variability within each meta-profile, the 40th and 60th percentile are reported in the associated metaprofiles.

**Analysis of population average single-cell ATAC-seq signal**. Single-cell ATAC-seq data for mixed *Drosophila* male and female 12 h embryos were obtained from Cusanovich et al.[50] (GEO dataset ID: GSE101581). The data matrix contains each individual cell on the column and the 2 kb binned genomic coordinates on the rows. We considered only those genomic regions corresponding to the chromosomes 2L, 2R, 3L, 3R or X.

The mixed population of cells were separated into two independent matrices corresponding to male and female embryos. To do this, for each cell a ratio was computed between the number of chrX reads and the number of reads found in all chromosomes (2L, 2R, 3L, 3R, X). Similarly to what described in the original study[50], cells with had a ratio ≤0.1625 were classified as male, whereas cells above this threshold were classified as female. Later, all cells which had less than a total of 500 counts were filtered out.

For each matrix (male or female), the total number of counts for each genomic region was taken and normalized by the library size (total number of counts in the cell matrix) of that region's chromosome. This was done, to account for the difference in copy number between chromosome X in male and female embryos. This procedure provided the library size normalized scATAC-seq counts. During further analyses, where specified the library size normalized scATAC-seq counts were converted to fold change over the chromosome average.

**MSL binding sites definition**. MSL binding site definitions were obtained from three previous studies. The refined list of high-affinity sites (HAS) were obtained from Ramirez et al.[8] Table S2 of the original article[8], and the original HAS list was obtained from Straub et al.[5] Table S1 of the original article[5]. CES sites were obtained from Alekseyenko et al.[4] Supplementary Table 1 of the original article. See also our Supplementary Data 3 for all regions mid-points.

**CLAMP binding sites definition**. CLAMP binding sites as defined by Soruco et al.[47] were provided by E. Larschan. See Supplementary Data 4.

**MSL and CLAMP binding sites around domain boundaries**. Mid-points of MSL and CLAMP binding sites were used as reference positions. For each factor ($m$), we computed the randomly expected binding sites per genomic bins ($E_m$) assuming a uniform distribution as null model: i.e. we divided the total number of binding sites ($N_m$) by the length of chrX ($L_X$) measured as number of (3.5 kb) bins.

$$E_m = \frac{N_m}{(L_X)} \qquad (11)$$

Next, for each domain border (belonging to the disappearing, appearing or same classes), we considered a window with size up to 15 bins (52.5 kb) on both sides. If such windows overlap for any pair of neighbouring domain boundaries, they are shortened by assigning equally to both boundaries the intervening region. This is an important point as avoids overestimating the association of any boundary class to genomic features, while allowing at the same time a definition of boundaries at fine scale (i.e. small domains).

Then we counted the number of binding sites windows around boundaries of each class, then divided by the windows total length. This result is the observed average number of binding sites per bins in the regions around boundaries of each class. The final results are reported as $\log_2$ ratio of observed over expected average number of binding sites per bin.

**Associating active gene TSS to their nearest boundary class**. The gene expression values were obtained from RNA-seq data for male S2 cells by Zhang et al.[34] (GEO GSE16344). ChrX genes were classified as dosage compensated (wild-type RPKM ≥ 4 and ratio of RPKM in MSL2 knockdown vs wild-type ≤ 0.74) or non-dosage compensated (wild-type RPKM ≥ 4 and ratio of RPKM in MSL2 knockdown vs wild-type > 0.74), similarly to what done by the original authors. Autosomal genes were classified as high expression (wild-type RPKM ≥ 4 and additional filter on ratio of RPKM in MSL2 knockdown vs wild-type > 0.74 to avoid confounding effects of facultative dosage compensation as described in Valsecchi et al.[64]), or low/no expression (wild-type RPKM < 4). When talking about active genes we consider the active genes in autosomes with the thresholds above together with non-dosage compensated (but active) genes on chromsome X (i.e. wild-type RPKM > 4 and ratio of RPKM in MSL2 knockdown vs wild-type > 0.74). See Supplementary Table 7 for the entire gene list.

For each TSS, the nearest domain boundary and the distance to the nearest boundary was found using the nearest and distancetoNearest functions from the GenomicRanges package[62], keeping only those genes which had a distance to

Nearest value < 10 kb. At this step, multiple genes may have the same nearest boundary. Therefore, to create a list of unique gene TSS and nearest domain boundary pairs, for each nearest boundary only the gene with the minimum distancetoNearest was considered. Furthermore, after this step all genes which were classified as low/no expression are removed from the list. The low or no expression genes were initially retained in the gene list to avoid misleading associations between high expression genes and boundaries in the autosomes.

**Distance of active genes to nearest domain boundary**. Gene were classified by expression patterns as described above. Around domain boundaries we considered a window of up to 15 bins, adjusting to avoid overlap with neighbouring boundary windows as described above. We then used the findOverlaps function from the GenomicRanges[62] package to compute the overlap between these windows and TSS of active genes (considering on both strands). Then we computed the distance between the TSS and the mid-point of the domain boundary.

**Processing of modENCODE DNase-seq data**. The modENCODE DNase-seq datasets for S2 cells and Kc167 cells were obtained directly from Dr. Peter Park lab as (samples ids 3324 and 3325, as detailed in (http://compbio.med.harvard.edu/modencode/webpage/Chromatin.v0.6.html), as the modENCODE portal has archived these datasets, thus FASTQ files are not directly available for download from the project data portal.

The DNase-seq reads were aligned to the dm3 genome considering only chromosomes 2L, 2R, 3L, 3R and X using bowtie version 1.1.2 as the FASTQ files contained old format Solexa quality scores. Bowtie alignment was run using the options: --mm to ensure memory-efficiency, -p 10 to run the bowtie alignment in parallel with 10 CPUs, --chunkmbs 200 was used to ensure a larger amount of memory allocation for each alignment. The alignment specific options were --sam, --best, --solexa-quals for solexa quality scores and -m 1 was used to return only reads which had a unique alignment.

Each of the two DNase-seq datasets contained three replicates. Coverage between replicates was similar between S2 and Kc samples, except for one replicate each on S2 and Kc samples. One replicate had very low coverage in S2 cells (8.6 M uniquely mapped reads vs 15.04 M average across the other two replicates), whilst another one had very high coverage in Kc cells (16.5 M uniquely mapped reads vs 15.3 M average across the other two replicates). We discarded these two replicates from further processing.

The processed BAM files for the replicates were imported into an R session using Rsamtools. The entire genome was then binned into 3.5 kb genomic intervals. The mapped reads were represented as a single bp mapping position to assign them to a unique genomic bin. The number of uniquely mapping reads per bin was counted and normalized by the total library size. The replicates were then merged for each sample by taking the average library size normalized DNase-seq counts per binned genomic interval.

During further analysis, when plotting the DNase-seq signal at domain boundaries near (distance ≤ 10 kb) active genes, the enrichment of DNase-seq signal was computed as the library size normalized DNase-seq counts at the boundaries divided by the average library size normalized DNase-seq counts in the region ±70 kb surrounding the domain boundaries. This signal was further transformed into the $z$-score to compare the S2 and Kc samples.

**Computing Insulation Score**. The insulation score as defined in Crane et al.[28] is calculated on our data as the mean Hi-C signal in a 35 kb (10 bins) squared sliding window. We started from our 3.5 kb Hi-C matrices and computed the insulation score moving the squared sliding window along the main diagonal. We ignored the first and last 10 bins of the chromosome. We removed the non-informative diagonals: first two diagonals, i.e. interactions occurring at distance 0 and 1.

The insulation score values were then normalized by the mean insulation score of each chromosome as in the original study[28]. Since the domain boundaries are defined at the intersection between the TAD start and end bins, the mean normalized insulation score from the two adjacent bins is considered.

**Distribution of normalized CLAMP signal files**. CLAMP ChIP-seq enrichment signal files (.wig files from GSE39271) were rescaled by dividing the signals in each file by their 99th percentile. This conservative normalization was applied to facilitate comparisons across samples accounting for potential differences in ChIP efficiency.

The CLAMP binding sites were assigned to the 3.5 kb genomic bin overlapping the mid-point of the binding site itself. For each bin containing a CLAMP binding site, the highest wig signal was obtained within the bin Start and End positions. This signal value was allocated as the probable CLAMP summit within that bin. We then fetched the unique list of nearest CLAMP bin (distance ≤ 10 kb) for each of the BEAF + and CP/CH + boundaries near to active genes and report the summit values for those bins.

**4C tag enrichment near domain boundaries**. The 4C data by Ramírez et al.[8] based on 18 probes were processed as is. The 4C data by Schauer et al.[9] based on 11 probes were instead further filtered as we noted larger differences between replicates for some probes. Namely, we discarded 4C data originating form a specific

probe if the two replicates have ≥ 2-fold difference in the total number of sequenced reads. To further avoid unbalanced comparisons, for each pair of samples compared (e.g. S2 WT vs MSL2-i) we considered a specific probe only if the it has ≤1.5-fold difference in the total number of sequenced reads across the compared samples. Thus, we obtained a total of 76 high quality 4C-seq dataset across 11 probes. See Supplementary Data 5 for the probe list and the corresponding GEO ids of all 144 experiments considered.

We used a similar strategy as Ramirez et al.[8]. First, we reassigned 4C read counts to our reference DpnII fragment ends table as obtained from the cooler package. Read counts per fragment were binarized thus assigning value of 1 to fragments with one or more overlapping reads, and a value of 0 to fragments without any overlapping read. Replicates are then further merged and converted to 1 or 0 values based on if a replicate contained any counts in the corresponding 4C-seq dataset.

To compute the 4C enrichment value ($E$), the fragments (frag) were further converted to their corresponding mid-points positions ($m$) and a small 20 kb window ($w_{small}$) was extended on both sides aggregating (by summing) all values ($v$) within that window. This sum was further divided by the sum of all values aggregated within a larger 600 kb window ($w_{big}$) used to estimate the expected background signal.

$$E_m = \log_{10}\left(\frac{\sum_{i=m-w_{small}}^{i=m+w_{small}} v_i}{\sum_{i=m-w_{big}}^{i=m+w_{big}} v_i} + 1\right) \tag{12}$$

With $w_{big} \leq m \leq \left(l_{chr} - w_{big}\right)$, where $l_{chr}$ is the length of chromosome (chr), to avoid windows extending beyond the chromosome start or end. The enrichment value $E$ constitutes the observed over expected 4C signal ratio and was $\log_{10}$ transformed with the addition of a pseudocount value of 1 for downstream analyses.

To summarize the average 4C enrichment signal around domain boundaries mid-points ($m_d$), grouped by class, the 4C data associated to fragments are mapped to the corresponding Hi-C bin ($b$) and their mean enrichment value assigned to the bin ($E_b$). A window up to distance $w$ from the boundary ($m_d$) is considered. The bins are then converted to their relative position ($p$) with respect to the bin containing the domain boundary mid ($b_{md}$). Thus, the enrichment values ($E_b$) are eventually assigned to their corresponding position ($E_p$) relative to any domain boundary (±35 kb). Finally, we compute the mean of enrichment values $E_p$ for each position ($p$).

**Reporting summary**. Further information on research design is available in the Nature Research Reporting Summary linked to this article.

## Data availability
All relevant data supporting the key findings of this study are available within the article and its Supplementary Information files or from the corresponding author upon reasonable request. All genomics data used in the paper have been previously released in public repositories as indicated in the paper or Methods details above. These include GEO datasets for Hi-C (GSE94115), MNAse-seq (GSE99893), ATAC-seq (GSE101581), RNA-seq (GSE16344), ChIP-seq (GSE39271) and modENCODE datasets for ChIP-chip (IDs 3745, 3748, 277, 908, 274, 925, 279, 3281, 5130, 5131, 5057, as detailed in the Methods section) and DNase-seq (IDs 3324 and 3325). The source data underlying Figs 1d, 2c–e, 3a, 3c, 4e, 5b, c, 5e, f, 6a, 6d, e and Supplementary Figs. 1c, 3a, 3c–f, 4d–f, 5a-c, 7b, 8d, and 9d are provided as a Source Data file. A reporting summary for this Article is available as a Supplementary Information file.

## Code availability
The R code for LSD domain calling procedure is available at https://bitbucket.org/koustavpal1988/fly_dc_structuralchanges_2018/

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

## Acknowledgements

We thank Pascal Carrivain for help on the polymer folding simulation models. We thank Peter Park and Youngsook Lucy Jung for help in retrieving archived modENCODE DNase-seq data. We thank Vincent Pirrotta and Yuri Schwartz for help in clarifying the annotations of modENCODE ChIP-chip datasets. We thank Marco Cosentino Lago-marsino, Peter Becker and Erica Larschan for critical feedback on earlier phases of this work. This work was supported by AIRC Start-up grant 2015 n.16841 to F.F.; AIRC postdoctoral fellowship to K.P.; ANR grant ANR-15-CE12-0006 EpiDevoMath, Fonda-tion pour la Recherche Médicale grant (DEI20151234396) and CNRS to G.C., D.J. and C.V., as well as computing resources by the Pôle Scientifique de Modélisation Numérique and Centre Blaise Pascal at ENS Lyon and CIMENT cluster at University Grenoble-Alpes.

## Author contributions

K.P. performed all analyses on genomics data. M.F. and E.S. collaborated with K.P. to analyses on insulators. D.J. and C.V. performed the polymer folding modelling. T.S. and G.C. obtained the sex-sorted fly embryos Hi-C data. E.M.C. and E.L. collaborated with K.P. to the single-cell ATAC-seq data analysis. F.F. coordinated the work. K.P. and F.F. wrote the paper.

## Competing interests

The authors declare no competing interests.
