## [Peer Review File · Nature Communications]

Reviewers' comments:

Reviewer #1 (Remarks to the Author):

The manuscript describes a study on the relation between dosage compensation and structural changes in drosophila chromosomes. In my opinion, this is a high-level study that sheds light upon a complex phenomenon and thus deserves publication on a high-level journal. I have just some minor comments that authors could address.

- 1) I am not sure I understand the model simulations. If an increase in contacts is observed for male chrX, isn't it straightforward that the change is not due to the homologous chromosome? Maybe this part should be explained better.
- 2) I find Fig. 4b of the Supp. Mat. misleading, as well as the associated discussion on true/false positive rates. The standard tad-calling algorithms are as arbitrary as the present one. I would avoid to refer to any tad-calling method as "true".
- 3) The authors report differences in structural domains between male and female in ChrX. Are there significant differences in other chromosomes? The change in scaling exponent does not seem to reveal changes, but this is not a very sensitive measure.
- 4) The observed correlation between upregulation of transcription and change in 3D structure of chromosomes is described in terms of upregulation causing structural change (first sentence of the results). Cannot it be vice versa?

Reviewer #2 (Remarks to the Author):

In this paper, the authors proposed several new computational methods to study the chromatin spatial organization in the Drosophila dosage compensated chromosome X. This work addresses an interesting research question. The major finding is the global conformational differences between male chrX and female chrX, which are associated with the gene expression difference in chrX. The paper is well written. Overall, I agree with the authors that tailored data analyzing methods are needed to study chrX chromatin organization. However, without further analyses and experimental validation, it is not clear that the conclusions made in the paper are fully supported by the results on real data. In my opinion, the manuscript is not ready for publication in Nature Communications in its current form. Here are my specific comments.

Major comments:

1. Page 5, paragraph 1, line 1. The major hypothesis is "global transcriptional up-regulation of chrX in the male fly is affecting its 3D structure". The authors imply the causality in such hypothesis. However, their data only support the "association" between 3D structure difference in chrX and transcriptional up-regulation. It would be more convincing if the authors can clearly demonstrate the direction of potential causal relationship. They can perform two additional experiments. (1) Disrupt chrX chromatin organization and measure gene expression changes, and (2) Alternate chrX gene expression and measure chromatin organization changes. These two experiments are necessary to clarify the causal relationship and improve the impact of this work.
2. Fig 1a contains many white dots where male chrX has more contacts than female chrX. Whether such difference is statistically significant and reproducible between biological replicates? Are those differentially interacting loci pairs associated with differentially expressed genes?
3. In Fig 2a, the authors called top 5% per diagonal as significant. It is unrealistic to assume that the number of significant interactions is a constant for each fixed genomic distance. The authors need to remove such assumption in interaction calling analysis. Since the authors analyzed high resolution Hi-C data, I would expect a comprehensive annotation of enhancer-promoter interactions in both male and female chrX. The authors need to develop tailored interaction callers,

and compare with two popular Hi-C interaction callers HiCCUPS and Fit-Hi-C.

4. Page 8, line -3. The authors conclude that "dosage compensated chrX is globally more accessible". However, more interactions do not necessarily indicate more accessible chromatin. For example, Lieberman-Aiden et al 2009 Science paper showed frequent interactions between compartment B (inaccessible chromatin). To validate this claim, the authors need to perform ATAC-seq experiments (e.g., PMID 29727464), and compare ATAC-seq peaks in male and female chrX.

5. The authors need to carefully benchmark their new TAD caller LSD. (1) What is the TAD boundary difference, when applying LSD to two biological replicates? In Fig 3C, they need to add the control: discordance between replicates. (2) Do they observed BEAF-32 or CP190 enriched in TAD boundaries? (3) Between male and female, whether dynamic BEAF-32/CP190 binding is correlated with dynamic TAD boundaries?

6. Fig 4a, male/female-specific TAD borders are NOT visually clear. Also some TAD borders are very close to each other, raising the question of robustness of proposed TAD caller LSD. It is better to show directionality index curve and insulation score.

7. To understand the consequence of induction or silence of DCC, the authors need to perform Hi-C experiments, instead of using 4C-seq data. It would be more informative to access the genome-wide of chromatin spatial organization differences, in addition to loci used in 4C-seq data.

Minor comments:

1. In Fig 1b, it is better to present male chrX and female chrX in the same figure to highlight their difference. Also, the log-log plot showed in Fig 1b is linear only locally. For example, the slope at the right edge (1Mb or above) is different from the slope in the middle. So the slopes listed in Fig 1c is not informative to capture the detailed pattern. It is better to fit the log-log curve by a piece-wise linear function.

2. The authors need to justify the validity of their polymer folding simulation, and demonstrate that their simulated Hi-C data is a close approximation of real Hi-C data.

3. Fig 2e shows a non-symmetric version of inter-chromosomal interaction map. It is better also show the ICE normalized inter-chromosomal interaction map, which is symmetric and easy to interpret.

4. Fig S4a. When calculating true positive rate and false discovery rate, how to define the "gold standard" of TAD boundaries?

Reviewer #3 (Remarks to the Author):

The main issue with the analysis (which the authors briefly touch on in the manuscript) is that without an allele specific setup it is impossible to differentiate between cis and trans contacts between homologous chromosomes. It is beyond the scope of this manuscript to generate these data but is a major confounding issue.

The authors state many assumptions that they can easily test but are not tested such as:

1) They state many times throughout the manuscript that the male X chromosome is expected to yield fewer reads (i.e. 50% less) than the two copy female X. They could determine this yield from

the data they have and see what the true distribution is and consider this in their down-sampling so more precise read balancing can be carried out.

2) They advocate for normalization methods that are geared toward copy number but I think this approach could actually end up biasing their analysis. It would be interesting to see how randomized non overlapping subsampling of the male Hi-C data would compare to one another.

3) For the non-parametric analysis, they set NAs, NaNs and infinite values to 0, this could be biasing their analysis. It would be nice to know what percentage of the data points this includes. Simply excluding such regions may be a better approach to avoid bias as non-parametric tend to approximate medians rather than means so adding enough 0s could be enough to skew interpretation.

4) Their polymer modeling does not seem to consider PCR/sequencing biases or the idea of chromosome territories (though it does consider pairing). I'm not sure how much weight these simulations actually contribute to their interpretations.

5) The fact that they don't observe a consistent decay in S2 cells (the other cell line tested clone 8 is thought to be essentially diploid) is problematic.

"We scored BG3-c2, Cl.8, D20-c2, D20-c5, D4-c1, L1, S1, W2, and D8 cell lines as minimally diploid, and S2-DRSC, S2R+, S3, Sg4, Kc167, D16-c3, and D17-c3 cell lines as minimally tetraploid. Our results for D9 and mbn2 cell line ploidy were inconclusive, due to the presence of multiple regions of relative read densities that were not ratios of whole numbers"

This shows that simply using methods that are not biased (in theory) for copy number may not be the best strategy when there are multiple X-chromosomes present (more noise) or there may be an issue with their imputation method for NAs/infinities.

6) Having just one X-chromosome is going to intrinsically be "higher resolution" for making inferences for that particular chromosome as the issue of not being able to differentiate trans homologous chromosomal interactions does not come into play though the issue of likely having a higher percentage of trans contacts should also be taken into account.

7) Instead of down sampling they should try to up-sample the male reads for the X so that there is less "noise".

8) What is the rationale for leaving out the Het regions or not using dm6?

We would like to thank the reviewers for their constructive feedback on our manuscript. We thoroughly revised our manuscript in light of these comments as detailed below.

To facilitate the reader in the point by point replies below we quote the original reviewers' comments in blue font and write our replies in black font, with indented text. In the revised manuscript we highlighted changes by using red font instead.

Reviewers' comments:

Reviewer #1 (Remarks to the Author):

The manuscript describes a study on the relation between dosage compensation and structural changes in drosophila chromosomes. In my opinion, this is a high-level study that sheds light upon a complex phenomenon and thus deserves publication on a high-level journal. I have just some minor comments that authors could address.

1) I am not sure I understand the model simulations. If an increase in contacts is observed for male chrX, isn't it straightforward that the change is not due to the homologous chromosome? Maybe this part should be explained better.

Based on this comment and similar requests for clarifications by Reviewer #3, we rephrased the motivation to further clarify the contribution of homologous chromosomes pairing to the observed interaction decay profile. We revised the text as follows:

"[...] since we are specifically focusing on the dosage compensated male chrX, we asked whether the lack of a physically proximal homologous chromosome may affect the distribution of Hi-C contacts. Indeed, the Hi-C read pairs originating from inter-chromosomal (trans) contacts between homologues are indistinguishable from intra-chromosomal (cis) contacts, thus potentially confounding the estimation of cis contacts. As the male chrX is the only chromosome without a paired homologue, we asked if this male chrX peculiarity may affect the observed differences in the interaction decay profile."

Moreover, after the first submission of our manuscript, a new work by Dr. Ting Wu's group presenting haplotype-resolved Hi-C data in first generation hybrids of different *Drosophila* strains became available, as it was publicly disseminated in 2 bioRxiv manuscripts (Abed *et al.*, bioRxiv <https://doi.org/10.1101/443887>; Erceg *et al.*, bioRxiv <https://doi.org/10.1101/443028>). Our original polymer folding simulations results seemed in line with results presented by Abed *et al.*, where the ratio of inter-homologous interactions vs cis-interactions is shown to increase at mid-/long-range distances (see Figure 2B in Abed *et al.*).

However, this more recent work by Wu's lab also clearly showed differences in distribution of "tight" vs "loose" homologous pairing across different chromatin states. As such, we decided to adopt a more sophisticated polymer folding simulations method incorporating the local epigenetic state of chromatin (Ghosh and Jost, PLoS Comput. Biol. 2018). We also revised the polymer folding simulations parameters to be in line with the frequency of "tight" pairing as estimated by Abed *et al.* in haplotype resolved Hi-C data. It's worth remarking that, despite the availability of experimental data with haplotype resolved Hi-C in *D. melanogaster*, the polymer folding simulations are still required to precisely disentangle the contribution of pairing to the cis-interaction profiles decay. Indeed, the simulations allow us to compare the interaction profiles obtained with or without pairing on chromosome X (chrX), the latter corresponding to the male genome.

The results of the new polymer folding simulations are now presented in Supplementary Figure 4. As discussed in the text, the more sophisticated polymer folding model considering chromatin states and the percentages of tight pairing described in Abed *et al.* shows that in fact the lack of pairing may result in a more flexible chromatin fibre, showing more long-range contacts. These increased long-range contacts due to a more flexible fibre are random events not resulting in

specific points of interactions. This is actually in line with our first manuscript version where we showed less clustered "top scoring interactions" in male chrX (Fig. 2a-c). We now show that the lack of pairing may partially explain the reduced clustering of "top scoring interactions" especially for longer range distances ($\geq 5\text{Mb}$), (Supplementary Fig. 3f). However, the pairing alone is not sufficient to explain the magnitude of differences observed in experimental data (Supplementary Fig. 3a-b), especially at shorter distances ($\leq 2\text{Mb}$) (Supplementary Fig. 3f).

The differences in "top scoring interactions" observed in experimental data can still be explained by a combination of lack of a paired homologous and locally more accessible chromatin in dosage compensated chrX. The local differences in chromatin accessibility and their association to differences in TADs separation are now also characterized more in details in the second part of our manuscript (also discussed in the replies to later points).

2) I find Fig. 4b of the Supp. Mat. misleading, as well as the associated discussion on true/false positive rates. The standard tad-calling algorithms are as arbitrary as the present one. I would avoid to refer to any tad-calling method as "true".

We apologize for the confusion caused by a mismatch between the legend text for the former Supplementary figure 4 (now Supplementary Fig. 5 in the revised manuscript) and the panels displayed in the figure itself. Unfortunately, this was due to a change introduced in the latest pre-submission draft of our manuscript, where we decided to remove a non-essential panel from the previous figure version to make room for the latest panel that was added in the submitted figure version. Unfortunately, the text legend in the uploaded supplementary file was still referring to the previous figure panels, thus having a shift compared to the displayed elements: panel "b" in the legend description was actually referring to panel "a" in the figure; "c" description was for figure panel "b" etc. Likewise, in the main text the reference to "Supplementary Fig. 4b" was actually describing the panel "a" of the same supplementary figure.

It turned out that the previously removed Supplementary Figure panel is indeed useful to address comment #6 by Reviewer 2, as such we have now re-introduced the panel (Supplementary Fig. 5a).

We apologize for these misleading descriptions caused by removal of a figure panel in the last pre-submission draft. We have now revised Supplementary Fig. 5 legends to correctly match the figure panels.

Then, for what concerns the reviewer's comment implying that there's a *de facto* lack of "true" TADs definition, we in general agree with that comment. This is due to the elusive biological definition of TADs for which there is not a consensus in the field. In fact, in our recent work on the comparative assessment of Hi-C data analysis methods (Forcato *et al.*, Nature methods 2017) to achieve a quantitative comparison of TAD calling algorithms we had only the option to generate simulated Hi-C contact matrices. In these simulated datasets, the location of simulated TAD borders is known, thus allowing to build a set of true positive and true negative TAD borders for algorithms benchmarking. While we acknowledge that simulated contact matrices may not have the complexity of real experimental Hi-C data, still the solution we adopted in (Forcato *et al.*, Nature methods 2017) up to now can be considered an acceptable practical compromise. Nevertheless, simulating datasets also allowed us to "add" increasing amounts of noise to the datasets, so as to compare the performances of different tools across an array of configurations with increasingly challenging conditions (increasing noise): see x-axis values in Supplementary Fig. 5b (corrected figure panel label).

The simulated datasets (from Forcato *et al.*, Nature methods 2017) have been used in Supplementary Fig. 5b to estimate the True Positive Rate (TPR) and False Discovery Rate (FDR) for the LSD algorithm presented in this manuscript. In Supplementary Fig. 4a, LSD performances are directly compared to the results obtained on the same simulated datasets by other methods

as already presented in (Forcato *et al.*, Nature methods 2017). As shown in Supplementary Fig. 5b, LSD has high TPR and low FDR comparable to the best performing methods. In fact, only TADBit has performances comparable to LSD across all levels of noise in the simulated datasets. However, TADBit has the additional limitation of having long data analysis time even for relatively low-resolution datasets (see Supplementary Figure 15 in Forcato *et al.*, Nature methods 2017). In our hands, running TADBit on human 10Kb resolution matrices was practically not feasible as it was taking several weeks to process the largest chromosomes (not shown). Whereas LSD can complete the same analysis in less than 30 minutes, thus allowing us to explore results obtained in a variety of experimental parameters and resolutions on data.

In one of the latest high-profile methodological publications in the field of Hi-C data analysis, the group of Dr. Marc Marti-Renom also adopted our definitions of true positives interactions (Spill *et al.*, Nature Communications 2019), thus confirming that the solutions adopted in our earlier work are still a reference in the field.

3) The authors report differences in structural domains between male and female in ChrX. Are there significant differences in other chromosomes? The change in scaling exponent does not seem to reveal changes, but this is not a very sensitive measure.

Based on methodological works on TAD calling algorithms by ourselves (Forcato *et al.*, Nature Methods 2017) and others (Dali and Blanchette, Nucleic Acids Research 2017), some level of variability is expected between structural domains borders calls even across experimental replicates. Indeed, in the first submitted version of our manuscript we showed that some variability in structural domains boundaries can be detected between male and female flies even when comparing autosomes (Fig. 3a,c and Supplementary Fig 5c,d). Nevertheless, the same figures clearly showed that the differences in structural domains borders between male and female are larger for chromosome X than for autosomes, independently of the TAD calling algorithm as well as the analysis parameters that are adopted.

In order to add further details on this point, we compared also the domain boundaries in male vs female autosomes, by grouping them in the "same", "disappearing" and "appearing" classes as previously defined for chromosome X. The results are summarized in the figure below, where the fraction of boundaries for each class ("same", "disappearing" and "appearing" in black, red, and green, respectively) are shown in the stacked barplot (y-axis for fraction of borders over their total number) for each chromosome (x-axis labels). As highlighted in this figure, "appearing" (green), "disappearing" (red) and "same" (black) classes contain comparable fractions of domain boundaries in each of the autosomes, whereas a specific difference in chrX is observed for the latter two classes: higher fraction of "disappearing" (red) and lower fraction of "same" (black) boundaries in chrX when comparing male vs female flies. This additional result is also in line with the observations on the weakening of several domain boundaries in male chrX in association to the binding of the dosage compensation complex (MSL complex), the local chromatin decompaction promoted by CLAMP, and the differences in insulators binding as presented and discussed in details in the revised manuscript.

For the sake of clarity, it's worth remarking that this figure has not been included in the current revised manuscript version as it was deemed not essential.

4) The observed correlation between upregulation of transcription and change in 3D structure of chromosomes is described in terms of upregulation causing structural change (first sentence of the results). Cannot it be vice versa?

We apologize for the misleading phrasing. We didn't mean to imply causality but to remark instead the association between the dosage compensation mechanisms and the structural changes. As such we revised the text as follows:

“To verify if the dosage compensated chrX in the male fly has a different 3D structure [...]”

Reviewer #2 (Remarks to the Author):

In this paper, the authors proposed several new computational methods to study the chromatin spatial organization in the *Drosophila* dosage compensated chromosome X. This work addresses an interesting research question. The major finding is the global conformational differences between male chrX and female chrX, which are associated with the gene expression difference in chrX. The paper is well written. Overall, I agree with the authors that tailored data analyzing methods are needed to study chrX chromatin organization. However, without further analyses and experimental validation, it is not clear that the conclusions made in the paper are fully supported by the results on real data. In my opinion, the manuscript is not ready for publication in Nature Communications in its current form. Here are my specific comments.

Major comments:

1. Page 5, paragraph 1, line 1. The major hypothesis is "global transcriptional up-regulation of chrX in the male fly is affecting its 3D structure". The authors imply the causality in such hypothesis. However, their data only support the "association" between 3D structure difference in chrX and transcriptional up-regulation. It would be more convincing if the authors can clearly demonstrate the direction of potential causal relationship. They can perform two additional experiments. (1) Disrupt chrX chromatin organization and measure gene expression changes, and (2) Alternate chrX gene expression and measure chromatin organization changes. These two experiments are necessary to clarify the causal relationship and improve the impact of this work.

As already mentioned about the comment #4 by Reviewer #1 we apologize for the misleading phrasing. We didn't mean to imply causality but to remark instead the association between the dosage compensation mechanisms and the structural changes. As such we revised the text as follow:

"To verify if the dosage compensated chrX in the male fly has a different 3D structure [...]"

Then, for what concerns the additional experiments suggested by this comment, we regret to say these points have been partially addressed in previous literature in this field, partially addressed in our work and partially are unfeasible.

Namely, for the first point about "disrupting chrX chromatin organization and measure gene expression changes" the reviewer is maybe suggesting to disrupt a specific TAD boundary and to verify the effect on transcription. However, 1) this would not really add information on the mechanisms related to dosage compensation; 2) it's already known in literature that disrupting a specific TAD border, thus inducing the merging of neighbouring TADs, does not necessarily result in activating silent genes or silencing active ones. This is sometimes referred to as "enhancer adoption". The mere exposure of a silent (or low expression) gene to an active enhancer by removing an intervening domain boundary does not change the activity status of the gene. This is supported by previous literature on the depletion of TAD boundaries by degrading cohesin (Rao et al., Cell 2107), where it is clearly reported that "loss of loop domains does not lead to widespread ectopic gene activation". This is also in line with other literature on the genetic editing of TADs, showing that "re-wiring of enhancer promoter interaction and aberrant [...] gene activation is not induced by a mere loss of insulation" at TAD borders (Despang et al., bioRxiv - <https://www.biorxiv.org/content/10.1101/566562v1>). In *Drosophila*, a distantly related work has taken advantage of balancer chromosomes to study gene expression changes upon multiple chromosome rearrangements induced by mutagenesis, finding small effects that can not be easily assigned to structural chromosomal alterations (Ghavi-Helm *et al.*, Nature Genetics 2019).

For the second suggestion about "altering chrX gene expression and measure chromatin organization changes", our understanding is that the reviewer is possibly suggesting either:

1) to silence (in males) or up-regulate (in females) the dosage compensation complex (DCC) so as to inhibit or induce the transcriptional up-regulation of chrX normally associated to the establishment of dosage compensation. However, this first solution has already been extensively explored in the field of *Drosophila* dosage compensation and we were as well reporting results based on MSL silencing in male cells (to inhibit the DCC activity) or Sxl silencing in female cells (to induce the DCC). In the novel manuscript version we have refined the results based on these experimental models now reported in (Supplementary Fig. 9c and 9e). Related to this point, additional data in support of the association between dosage compensation mechanisms (DCC or its cofactor CLAMP) and chromatin structure have been added in the revised (Fig. 5 and Supplementary Fig. 9).

2) or to induce expression of one or more genes with a drug or other treatment, then verifying if that results in an increase chromatin accessibility and weakening of TAD borders. However, the peculiarity of fly dosage compensation is that the transcriptional up-regulation can't be disjointed from the epigenetics marks changes (H4K16 acetylation). Indeed, in fly dosage compensation it can't be ultimately established if the "increased accessibility", as already documented in literature (Urban et al., PLoS One 2017), is definitely due only to epigenetic changes (H4K16ac deposition) or only to transcription up-regulation or to CLAMP action alone, as the three things happen concurrently and are at least partially dependent on each other.

Thus, in reply to this point we have re-phrased the misleading statement about the causal connection. Instead we refined and extended the data in support of the association between dosage compensation mechanisms and chromatin structure in (Fig. 5 and Supplementary Fig. 9).

2. Fig 1a contains many white dots where male chrX has more contacts than female chrX. Whether such difference is statistically significant and reproducible between biological replicates? Are those differentially interacting loci pairs associated with differentially expressed genes?

The data shown in Fig. 1a can be considered as a local snapshot for the general trend shown in figures reporting the interaction frequencies slope decay with distance, where indeed the slope of male chrX is less negative than other chromosomes (Fig. 1b and related Supplementary Figures 1 and 2).

In order to make Fig. 1a more directly comparable to the plots with interactions decay, in the revised version of our manuscript we replaced the original Fig. 1a, that was based on the log₂ ratio of normalized Hi-C counts in male over female sex sorted embryos. The revised figure is instead based on the same log ratio, but comparing normalized Hi-C contact frequencies scaled by the average contact frequencies in the main diagonal (see methods). It must be noted that the current cooler library version also implements a similar approach by default wherein the bias factor is rescaled by the square root of the mean bias factor (Abdenour and Mirny, bioRxiv 2019 - <http://dx.doi.org/10.1101/557660>). We used the cooler library ourselves, but an earlier version where this was not implemented yet

We'd also like to remark that in the first submitted version of our manuscript we already showed that the biological replicates are indeed showing the same pattern of interactions decay (Supplementary Fig. 1a). In order to address more directly the request by this reviewer we are showing here the pairwise log ratios of normalized Hi-C contact frequencies between each male and each female samples of sex sorted embryos (total 4 pairs).

For the sake of clarity, we remark that we are not including these figures in current revised version of the manuscript, as we argue that the reproducibility of the interactions decay pattern across replicates has already been addressed in (Supplementary Fig. 1a). Furthermore, in the revised version of our manuscript we have assessed the reproducibility of replicates by computing the stratum-adjusted correlation coefficient (SCC) with the hicrep tool (Yang et al., Genome Research 2017). The sex sorted embryos replicates show an SCC value of 0.97 for the male and 0.98 for the female samples. The SCC has been computed up to a 2.5Kb distance on matrices binned at 25Kb bin size. We revised the text to include these results as follow:

"In most of the analyses, the biological replicates of male or female sex sorted embryos were merged, as we verified their reproducibility by computing the stratum-adjusted correlation coefficient (SCC) with the hicrep tool⁵⁸. The sex sorted embryos replicates show an SCC value of 0.97 for the male and 0.98 for the female samples. The final read numbers are available in Supplementary Table 1."

For what concerns the final remark about "differentially interacting loci pairs associated with differentially expressed genes", we must remark that we do not claim these to be differentially interacting loci. As discussed in (Fig. 1b, c, d) and related (Supplementary Figures 1 and 2), we do test the significance in the global differences in trends of interactions decay with distance, but we do not claim each of the points with relatively higher signal in male as significant differences. This would be problematic from a methodological point of view for the underlying copy number difference resulting in unreliable differential interaction calls, as discussed later in our manuscript as well.

Moreover, based on our analyses on the clustering of top-scoring interactions (Figure 2a,b,c and related Supplementary Fig. 3a,b,c) and on the higher prevalence of trans-interactions (Figure 2d,e), we conclude that the relatively higher interaction frequencies in mid-long range distances are not due to stable loops between specific genomic loci. Instead, we claim they are random contacts arising partly due to a general higher chromatin accessibility in male chrX (see also data on weakening domain borders) and partly due to more flexibility of the unpaired male chrX (see Supplementary Fig. 3f and Supplementary Fig. 4 for data on simulations).

3. In Fig 2a, the authors called top 5% per diagonal as significant. It is unrealistic to assume that the number of significant interactions is a constant for each fixed genomic distance. The authors need to remove such assumption in interaction calling analysis. Since the authors analyzed high resolution Hi-C data, I would expect a comprehensive annotation of enhancer-promoter interactions in both male and female chrX. The authors need to develop tailored interaction callers, and compare with two popular Hi-C interaction callers HiCCUPS and Fit-Hi-C.

Based on this comment we understand that maybe there was a misunderstanding on the rationale behind our analysis of clustering of top-scoring interactions. In fact, we do not claim these to be "significant" contacts. We agree with the reviewer that a fixed percentage of normalized contact frequencies would not represent significant "loops" of interaction. Our approach stems from the observation that higher coverage yields an exponential increase in the statistical power to detect interactions (Forcato et al., Nature Methods 2017). As such, the 2 copies of female chrX, which is also the sample with a larger number of filtered reads (Supplementary Table 1) will always result in a larger number of interactions being called in the female.

Instead, our approach selects a fixed number (percentage) of top scoring interactions per each chromosome, then it focuses on assessing if they are differentially clustered across samples and chromosomes. The focus on the level of clustering of top scoring interactions is based on the expectation that a sample with more stable localized contacts would yield larger clusters of top scoring interactions. Instead, the general lower level of top scoring interactions clustering observed in male chrX (Fig. 2c), coupled with the less steep interactions decay with distance (Fig. 1b) and the results of polymer folding simulations (Supplementary Fig. 4), support the conclusion that random contacts occurring at mid-/long- range distances in male chrX, as a consequence of higher accessibility and flexibility.

Our approach focused on the clustering of top scoring interactions is not calling loops. Thus, it is not comparable with interaction callers such as HiCCUPS and Fit-Hi-C. Moreover, as we showed in (Forcato et al., Nature Methods 2017), these methods yield very variable results, as well as any other interaction calling algorithms. This tools benchmarking is hampered by the lack of a consensus reference gold standard set of true interactions.

To clarify the rationale behind our approach, and how it is different from loop calling, we revised the text as follow:

"We aimed to clarify if the higher interaction signal at mid- and long-range distances reflects a difference in specific point interactions. A first approach could be based on comparing the significant point interactions identified in the male and female Hi-C data matrices. However, as we observed in our recent study³⁶, all algorithms for detecting point interactions in Hi-C show a strong dependency between the number of called interactions and the coverage (number of reads). As the single copy male chrX yields relatively fewer reads than the dual copy female chrX, this is expected to introduce a bias in the number of point interactions. To overcome this limitation, we devised a non-parametric approach to compare the pattern of interactions (see Methods). This procedure is based on selecting the highest scoring pairwise interactions by applying a threshold on quantiles computed independently for each diagonal of intra-chromosomal normalized Hi-C data matrices (Fig. 2a). In our approach, the resulting "top-scoring" interactions are by definition a fixed number of data points. Thus, they should not be considered as significant point interactions per se. Instead, we focus on assessing if they are differentially distributed across samples and chromosomes."

4. Page 8, line -3. The authors conclude that "dosage compensated chrX is globally more accessible". However, more interactions do not necessarily indicate more accessible chromatin. For example, Lieberman-Aiden et al 2009 Science paper showed frequent interactions between compartment B (inaccessible chromatin). To validate this claim, the authors need to perform ATAC-seq experiments (e.g., PMID 29727464), and compare ATAC-seq peaks in male and female chrX.

We agree with the reviewer's comment that "more interactions" do not necessarily indicate more accessible chromatin. However, we'd like to remark that:

- 1) the "compartment" analysis on Hi-C data to identify active ("A") and inactive ("B") compartments as described in (Lieberman-Aiden et al., Science 2009) is an analysis focused on large scale patterns, as indeed data are generally summarized with very large genomic bins to visualize the "checkered" pattern associated to compartments. In the original (Lieberman-Aiden et al., Science 2009) publication the data were summarized with 1Mb size bins, as the very first Hi-C dataset had low coverage. However, even with high coverage data compartments are expected to be apparent when summarizing data at lower resolution, as shown for example in (Rao et al., Cell 2014; Fraser et al., Microbiol. Mol. Biol. Rev. 2015). This is due to the fact that the checkered pattern of compartments interaction becomes apparent at larger distances, where the coverage summarized over smaller genomic bins would be very low.
- 2) instead, the relatively higher contact frequencies in male chrX are observed at mid-/long-range distances (especially in the 500Kb-1Mb range, as shown in Figure 1), thus at a much shorter scale compared to compartments interactions.
- 3) the latest literature on Hi-C performed on synchronized mitotic cells support as well the observation that compacted prometaphase chromosomes will yield an Hi-C contact maps with contacts localized at short distance (Gibcus et al., Science 2018).
- 4) The generally higher accessibility of dosage compensated chrX is expected based on established literature in the field, as H4K16ac is deposited on male chrX by the dosage compensation complex (DCC) and this histone modification is expected to increase the accessibility of chromatin (Gelbart et al., Nat Struct Mol Biol 2009; Shogren-Knaak et al., Science 2006; Bell et al., Nat Struct Mol Biol 2010;).
- 5) Also more recent literature in the field of dosage compensation supports the role of specific DCC co-factors in promoting chromatin accessibility (Urban et al., PLoS One 2017).
- 6) Our conclusions are better focused in the revised version of the manuscript, based on the combined observations on less steep interactions decay with distance (Fig. 1b), coupled with a lower level of top scoring interactions clustering observed in male chrX (Fig. 2c), coupled with the polymer folding simulation results suggesting a more flexible fiber for the unpaired chrX (Supplementary Fig. 4). These observations support the hypothesis of more random interactions occurring at mid-/long-range distances in male chrX, as a consequence of higher accessibility and flexibility.

Nevertheless, to further elucidate the differences in chromatin accessibility, as suggested by the reviewer we included additional analyses using MNase-seq, ATAC-seq and DHS data in the revised manuscript.

Each of the three techniques provide different resolutions for chromatin accessibility. Primarily, MNase-seq signal provides broader peaks, whereas ATAC-seq and DHS signal yields sharper peaks. Using the new stratification of domain boundaries based on combinatorial binding of insulators (discussed below and in the revised manuscript), we observe that accessibility as quantified by MNase-seq (Urban et al., PLoS One 2017) is generally higher for S2 cells (male) near all three classes of domain boundaries (Fig. 5a and Supplementary Fig. 9a). Furthermore, it is only the S2 cells which show a significant change in median chromatin accessibility upon CLAMP knock-down (Fig. 5b). No changes were observed upon MSL2 knock-down. This observation agrees with prior findings in literature wherein CLAMP was implicated in mediating changes in chromatin accessibility in dosage compensated chrX.

We also noticed that active gene transcription start site (TSS) tend to be near boundaries containing both BEAF32 and CP190 or Chromator (Fig. 5d). We obtained recently generated single-cell ATAC-seq data on male and female embryos (Cusanovich et al., Nature 2018). Using this dataset, we created population level chromatin accessibility maps unbiased by copy number for both male and female embryos. Observing gene TSSs near each class of domain boundaries (less than 10Kb distance), we found that active gene TSSs tend to stratify differently when classified by their nearest domain boundaries. TSSs which tend to be near boundaries containing low signal for BEAF32 and CP190 or Chromator showcase very low accessibility, TSSs near boundaries containing high signal for either one protein showcase accessibility near the middle of the distribution. Finally, TSSs near boundaries containing high signal for both BEAF32 and CP190 or Chromator showcase high accessibility. This is confirmed across both autosomes and chromosome X. Yet, only chrX showcases significantly different accessibility between male and female embryos.

We also obtained the modENCODE DNase-seq data for Kc167 and S2 cell lines and we confirmed the same findings observed in single cell ATAC-seq data.

5. The authors need to carefully benchmark their new TAD caller LSD. (1) What is the TAD boundary difference, when applying LSD to two biological replicates? In Fig 3C, they need to add the control: discordance between replicates. (2) Do they observed BEAF-32 or CP190 enriched in TAD boundaries? (3) Between male and male, whether dynamic BEAF-32/CP190 binding is correlated with dynamic TAD boundaries?

We would like to remark that in the first version of our manuscript we included the results of LSD benchmarking against other TAD callers on simulated datasets in (now in Supplementary Fig. 5b). As discussed in the reply to a comment by Reviewer #1 there was an unfortunate shift in the figure legends due to the late removal of a figure panel. We have now amended the Supplementary Figure legend to avoid any confusion. Nevertheless, the results reported in Supplementary Fig. 5b show the benchmarking on the same simulated datasets generated for our recent work on the comparative assessment of Hi-C data analysis methods (Forcato *et al.*, Nature methods 2017). LSD has high True Positive Rate (TPR) and low False Discovery Rate (FDR) comparable to the best performing methods. In fact, only TADBit has performances comparable to LSD across all levels of noise in the simulated datasets. However, TADBit has the additional limitation of having long data analysis time even for relatively low-resolution datasets (see Supplementary Figure 15 in Forcato *et al.*, Nature methods 2017). In our hands, running TADBit on human 10Kb resolution matrices has been practically impossible as it was taking several weeks to process the largest chromosomes (not shown). Whereas LSD can complete the same analysis in less than 30 minutes, thus allowing us to explore results obtained in a variety of experimental parameters and resolutions on data.

In addition to this benchmarking on simulated datasets, as suggested by the reviewer, we are here reporting the concordance of TAD boundaries when applying LSD on two biological replicates for the male and female embryos dataset, respectively. As reported in the figure below, in general the TAD boundaries are highly concordant across replicates analyzed with lower DI-window parameters (DI window 5 or 10 in the figure below), thus confirming the analysis of these data at high resolution yields robust results. We must note that both male and female replicates show high level of concordance across the whole range of possible DI-window parameters for autosomes and double copy female chrX as well, thus confirming the software itself is yielding robust results. The single copy male chrX has higher variability at larger DI-window parameters. This may be expected based on the fact that this is a 10Kb bins size contact matrix. As such, the coverage is relatively low at larger distances, especially in the single copy chromosomes, thus the larger DI-windows would capture a noisier part of the contact matrix. To further confirm the concordance between replicates we computed the stratum adjusted correlation coefficient (SCC) (Yang et al., Genome Research 2017). The SCC value is 0.97 for the male embryos and 0.98 for the female embryos replicates.

Then, for what concerns the question #2 in this reviewer's comment, we did in fact assess the presence of both BEAF-32 and CP190 at the TAD borders. These data were reported in the Supplementary Fig. 5, showing that binding peaks for both insulators are found at TAD borders, with higher frequency at "disappearing" and "same" classes of boundaries (Supplementary Fig. 5c). This was also confirmed by the ChIP enrichment signal (Supplementary Fig. 5d). The enrichment of CTCF is instead less pronounced, as if is expected based on previous literature on in *Drosophila* TAD borders (Sexton et al, Cell 2012).

In the revised manuscript we extended this analysis by adding also information on the binding of Chromator (Supplementary Fig. 7).

Also related to these data and to the question #3 in this reviewer's comment, in the revised manuscript we further investigated the contribution of "dynamic" (i.e. variable levels) of insulators binding in relation to dynamic TAD boundaries. Namely, we adopted a new strategy for the stratification of domain boundaries based on combinatorial binding of insulators. We regrouped domain boundaries based on the presence and level of enrichment in binding of BEAF32 and (CP190 or Chromator). This grouping of insulators is motivated by earlier literature reporting that BEAF32 engage in long-range interactions in cooperation with CP190 or Chromator. Thus we group domain borders with either one of the latter two insulators bound. We further consider their level of enrichment (high or low), to define 3 classes of domain boundaries: "BEAF+ and CP/CH+"; "BEAF- and CP/CH-"; "BEAF+ or CP/CH+". As extensively discussed in the revised manuscript, this alternative stratification allowed us to refine the characterization of male-specific differences in dosage compensated chrX.

6. Fig 4a, male/female-specific TAD borders are NOT visually clear. Also some TAD borders are very close to each other, raising the question of robustness of proposed TAD caller LSD. It is better to show directionality index curve and insulation score.

As suggested by the reviewer we revised the figure by adding both a line-plot showing the insulation score ratio between male and female samples, and the lines showing the directionality index for the male and female samples, separately.

In our analyses we considered the effect of merging or not the TAD borders that appear to be very close to each other. However, we found that this has not a significant effect on the downstream analyses. For example, if we consider the different enrichment of MSL binding sites around "appearing", "disappearing" and "same" classes of boundaries, when merging TAD borders separated by just one bin size we still obtain a similar result as shown in the figure below.

In the final version of our results, we preferred to report results without merging nearby TAD borders because 1) the bin size is still larger than the average *Drosophila* gene size 2) the resolution of other genomics dataset is also higher. To this concern, as shown also in Supplementary Fig. 5a adjacent bins in Hi-C data may correspond to distinct peaks of insulators. Thus, in general we didn't want to reduce the resolution of the results before the downstream analyses.

7. To understand the consequence of induction or silence of DCC, the authors need to perform Hi-C experiments, instead of using 4C-seq data. It would be more informative to access the genome-wide of chromatin spatial organization differences, in addition to loci used in 4C-seq data.

We must note that this type of experiment has already been done in the field of *Drosophila* dosage compensation by Ramirez et al, Mol Cell 2015. In this article the authors performed Hi-C in S2 cells in control and MSL2 or MSL3 RNAi samples. The authors reported that they could not detect large scale changes in TAD structure based on Hi-C data, whereas they could detect more local specific changes in interactions based on 4C data with higher although local resolution. For this reason, we initially focused on the 4C data to obtain an independent confirmation of our findings.

As suggested by the reviewer we reanalyzed this Hi-C dataset as well. As reported in the figure below, showing the pairwise Spearman correlation between the insulation score vectors, we confirm that the insulation score has minor changes after the silencing of MSL2 or MSL3, but these changes are specifically larger for male chrX (lower correlation) rather than autosomes. For the sake of clarity, we are not including this figure in the revised manuscript as the 4C data that we originally included, as well as the additional data on DNase-seq, MNase-seq and scATAC-seq provide information at a higher resolution.

In the revised manuscript we also clarified that the main differences between domain boundaries in male and females are explained by differences in accessibility rather than 3D interactions. CLAMP and chromatin insulators (BEAF-32, CP190 and Chromator) are the features more clearly explaining the observed differences.

Minor comments:

1. In Fig 1b, it is better to present male chrX and female chrX in the same figure to highlight their difference. Also, the log-log plot showed in Fig 1b is linear only locally. For example, the slope at the right edge (1Mb or above) is different from the slope in the middle. So the slopes listed in Fig 1c is not informative to capture the detailed pattern. It is better to fit the log-log curve by a piece-wise linear function.

As requested by the reviewer we are here reporting the log-log plot as in Figure 1b but with male and female chrX together in the same plot (right plot), along with a representative autosome chromosome arm (3R, left plot).

This plot is reported to confirm that the differences between male and female chrX are highlighted also in this version of plot, as expected by the reviewer. However, in the revised manuscript we are keeping the original settings in which chrX is compared to autosomes within the same sample, rather than across samples. This is a more conservative solution as minor differences across samples (in terms of coverage) or differences across datasets (in terms of minor protocol variations) may affect the data.

For what concerns the slope coefficients, we actually computed the linear model only considering distances <400Kb (≤ 5.6 in log₁₀ scale), as indicated in the manuscript. The rationale behind this

choice is indeed to capture the slope decay in its almost linear portion, before interaction decay curve becomes noisier at larger distances, as pointed out also by the reviewer.

We also would like to remark that larger distances interactions are instead considered in the alternative Kuiper's statistics computed on the cumulative density functions (CDFs) of the interaction probability decay plots. In this alternative statistic, that is confirming the differences observed by comparing slopes (Supplementary Fig. 1c), we considered distances up to 2.5Mb as indicated in the manuscript text.

To clarify this point we revised the manuscript text as follows:

"Finally, we confirmed that the observed differences are significant by comparing either the slope coefficients deltas (Wilcoxon test p -value < 0.01) (Fig. 1c) or the cumulative density functions of contact frequencies, which are independent of parameters estimation on the decay curves (Kuiper's statistic – see Methods) (Supplementary Fig. 1d)."

2. The authors need to justify the validity of their polymer folding simulation, and demonstrate that their simulated Hi-C data is a close approximation of real Hi-C data.

As discussed more extensively in the replies to specific comments by Reviewers #1 and #3, in the revised manuscript we have extended and revised the polymer folding simulations. The new results are based on a more advanced model considering chromatin states and the chromosomes pairing parameters inferred by experimental data. These results are discussed in details in the text and presented in (Supplementary Figure 4).

3. Fig 2e shows a non-symmetric version of inter-chromosomal interaction map. It is better also show the ICE normalized inter-chromosomal interaction map, which is symmetric and easy to interpret.

Based on this comment we realized that maybe there was a misunderstanding on the content and purpose of the analysis presented in Fig 2e. Namely, this figure is not actually reporting an inter-chromosomal contact map. Instead it is reporting a summary of the relative distribution of trans (inter-chromosomal) read pairs originating from each autosome arm and chrX. This relative distribution is reported as a chromosome arm level summary of observed over expected reads distribution.

To avoid such misunderstandings, we rephrased the description of these results as follow:

"A more accessible chrX is expected to yield more inter-chromosomal Hi-C contacts. Thus, we assessed the total inter-chromosomal (trans) contacts, defined as read pairs with either end mapped in different chromosomes, compared to the total intra-chromosomal (cis) contacts. We noted that trans contacts are relatively more frequent specifically on male chrX, as confirmed both in sex-sorted embryos and cell lines datasets (Fig. 2d, and Supplementary Fig. 3d). This is compatible with a globally more accessible male chrX. However, we reasoned that this might also be due to a technicality of Hi-C data analysis, as trans contacts between homologous chromosomes are indistinguishable from intra-chromosomal contacts. As male chrX is the only chromosome without a homologous one in our dataset, the relatively higher number of trans contacts might just be due to the absence of confounding contacts between homologues. To rule out this possibility, we examined the distribution of trans contacts across chromosomes and verified that in sex-sorted embryos trans contacts originating from each single male autosome are specifically enriched in chrX (Fig. 2e). This pattern is not compatible with a confounding effect due to the single copy of chrX and can only be explained by chrX indeed engaging in more trans contacts with all other chromosomes."

4. Fig S4a. When calculating true positive rate and false discovery rate, how to define the "gold standard" of TAD boundaries?

This point is also related to a similar comment by Reviewer #1 above. As detailed above, there's a *de facto* lack of "true" TADs definition. This is due to the elusive biological definition of TADs for which there is not a consensus in the field. In fact, in our recent work on the comparative assessment of Hi-C data analysis methods (Forcato *et al.*, Nature methods 2017) to achieve a quantitative comparison of TAD calling algorithms we had only the option to generate simulated Hi-C contact matrices. In these simulated datasets, the location of simulated TAD borders is known, thus allowing to build a set of true positive and true negative TAD borders for algorithms benchmarking. While we acknowledge that simulated contact matrices may not have the complexity of real experimental Hi-C data, still the solution we adopted in (Forcato *et al.*, Nature methods 2017) up to now can be considered an acceptable practical compromise. Nevertheless, simulating datasets also allowed us to "add" increasing amounts of noise to the datasets, so as to compare the performances of different tools across an array of configurations with increasingly challenging conditions (increasing noise): see x-axis values in (Supplementary Fig. 5b - updated figure panel label).

The simulated datasets (from Forcato *et al.*, Nature methods 2017) have been used in Supplementary Fig. 5b to estimate the True Positive Rate (TPR) and False Discovery Rate (FDR) for the LSD algorithm presented in this manuscript. In Supplementary Fig. 4a, LSD performances are directly compared to the results obtained on the same simulated datasets by other methods as already presented in (Forcato *et al.*, Nature methods 2017). As shown in Supplementary Fig. 5b, LSD has high TPR and low FDR comparable to the best performing methods. In fact, only TADBit has performances comparable to LSD across all levels of noise in the simulated datasets. However, TADBit has the additional limitation of having long data analysis time even for relatively low-resolution datasets (see Supplementary Figure 15 in Forcato *et al.*, Nature methods 2017). In our hands, running TADBit on human 10Kb resolution matrices has been practically impossible as it was taking several weeks to process the largest chromosomes (not shown). Whereas LSD can complete the same analysis in less than 30 minutes, thus allowing us to explore results obtained in a variety of experimental parameters and resolutions on data.

Reviewer #3 (Remarks to the Author):

The main issue with the analysis (which the authors briefly touch on in the manuscript) is that without an allele specific setup it is impossible to differentiate between cis and trans contacts between homologous chromosomes. It is beyond the scope of this manuscript to generate these data but is a major confounding issue.

We acknowledge and agree with the reviewer's opinion that generating additional allele-specific data to investigate contacts between homologous chromosomes would be beyond the scope of the manuscript. However, as discussed in the reply to the first comment by Reviewer #1, haplotype resolved Hi-C data for *Drosophila melanogaster* became available from an independent manuscript pre-print published on bioRxiv (Abed *et al.*, bioRxiv <https://doi.org/10.1101/443887>). This manuscript allowed us 1) to confirm that the ratio *trans-homologous* over *cis-interactions* is relatively higher at mid-long distances, and this was in line with our previous polymer folding simulations; 2) to infer more precise parameters on the distribution of tight homologous chromosomes pairing and its preferential association with specific chromatin states, so as to incorporate them in a more sophisticated polymer folding simulation model. The results about this part of the manuscript have been expanded and allow us to refine the conclusions to this concern. In particular, the lack of a homologous chromosome is associated to increased flexibility of the polymer, resulting in more random interactions especially at mid-/long-range distances, whereas other local differences in interactions and structural domains are correlated to the localization of the dosage compensation complex, its co-factor CLAMP and associated chromatin insulators.

The authors state many assumptions that they can easily test but are not tested such as:

1) They state many times throughout the manuscript that the male X chromosome is expected to yield fewer reads (i.e. 50% less) than the two-copy female X. They could determine this yield from the data they have and see what the true distribution is and consider this in their down-sampling so more precise read balancing can be carried out.

Based on the reviewer's comment we understand that maybe some details on how the different coverages were taken into account in specific analyses were not clear.

It's worth remarking that, in order to disentangle the potential confounding effect of different coverage between autosomes and chromosome X, in the analysis of the interactions decay with distance we also showed the results are the same if obtained after down-sampling autosomes read to 50% (Supplementary Fig. 1b). In this case we were comparing slopes between chromosomes within the same Hi-C sample of sex sorted embryos, which do not have aneuploidies as would be the case in cell lines. We do not expect the ratio of reads between chrX and autosomes to be 0.5 as they differ also in length, and the coverage do not increase linearly with length as Hi-C measures pairwise contacts. However, we can reasonably assume that 1 copy of each individual autosome could be approximated by 50% of cis-reads measured in the diploid genome for that autosome. This was the rationale behind using 50% of male autosomes reads when comparing their interaction slope decay against the single copy male chromosome X in (Supplementary Fig. 1b)

Instead, when comparing the differences in clustered "top scoring" interactions between male and female, we were dealing with an inter-sample comparison (Supplementary Fig. 3a). In this case we computed for each chromosome the ratio of coverage in male over female sample, then used this ratio to downsample the female reads to the same number of reads observed in the male. In the sex sorted fly embryos dataset the male has in general a lower coverage than the female sample (37% lower on average in the counts of filtered reads for each autosome). The exact subsampling ratios used for each chromosome are reported in the table below. As the tables is showing, the ratio for chromosome X is exactly 50% of the average ratios for the other

autosomes (0.315 for chrX vs average of 0.63 on autosomes), thus confirming that also the assumption of 50% downsampling for intra-sample comparisons as discussed above was correct.

chr2L	chr2R	chr3L	chr3R	chrX
0.6297883	0.6249765	0.6356459	0.6221048	0.3150378

For improved clarity we have now included these numbers and the absolute coverage for each dataset, sample and chromosome in the revised manuscript as Supplementary Table 8. The main text paragraphs referring to down-samplings are the following ones:

In the Results section:

"This observation is concordant across replicates (Supplementary Fig. 1a) and not dependent on the lower male chrX coverage, as expected due to copy number differences between sexes (Supplementary Fig. 1b - see Methods)."

and also

"We also considered if the less clustered distribution of top-scoring interactions in the single copy male chrX may be due to lower coverage yielding noisier data. To further clarify this point, we repeated the analysis on Hi-C contact matrices obtained by down-sampling read pairs for each female chromosome to the read count of the corresponding male chromosome (see Methods). The analysis of clustered top-scoring interactions confirms the same pattern, which is still present after reads down-sampling (Supplementary Fig. 3a - Downsampling)"

Then in the Methods section:

"To account for the disparities in coverage due to copy number and sequencing depth differences between male and female samples, we used two approaches for down sampling of read counts, as indicated in the results. In the first, we simulated the effect of single copy number on male autosomes by randomly down sampling 50% of the autosomal reads in the male samples. For this we used the `numpy.random.binomial` function in python with the probability parameter set to 0.5. The down-sampled observed read counts were then normalized using chromosome-wise ICE. This method was used to check the rate of Hi-C decay when the copy number of autosomes is similar to that of chrX (Supplementary Fig. 1b).

In the second approach, we down-sampled all female chromosomes (observed cis read counts) by the ratio of cis interactions count present in the corresponding male chromosome (Supplementary Table 8), to make the total sum of observed cis read counts comparable between male and female samples, chromosome by chromosome. The down-sampled observed read counts were normalized with chromosome-wise ICE. This approach was used to verify the effect on TAD calls, and the effect on clustering of top-scoring interactions between the male and down-sampled female samples (Supplementary Fig. 5d and Supplementary Fig. 3a, respectively)"

2) They advocate for normalization methods that are geared toward copy number but I think this approach could actually end up biasing their analysis. It would be interesting to see how randomized non overlapping subsampling of the male Hi-C data would compare to one another.

As mentioned in the previous point, we adopted different strategies of reads subsampling (for intra-sample or inter-sample comparisons) to confirm the difference in copy number and coverage alone are not sufficient to explain the observed differences in the interaction profiles.

Namely, in Supplementary Fig. 1b we subsampled reads from each autosome arm to match the read count for chrX in male. The results are quite robust as repeating the procedure over 100 iterations of reads subsampling without replacement, as suggested by the reviewer, then performing the whole procedure including data re-normalization and interaction slope decay computation yields the results reported in the figure below. In the left panel we show the average interaction slope decay across all 100 iterations along with its 95% confidence interval based on the 100 iterations (not visible in the plot because the confidence interval is very narrow). In the right boxplot we show the distribution of slope coefficients differences computed for each chromosome over each iteration similarly to what done in the previous Supplementary Fig. 1b.

We are currently not including this figure in the revised manuscript as it was deemed not essential and not adding relevant information as the span of the confidence interval on iteration is negligible.

3) For the non-parametric analysis, they set NAs, NaNs and infinite values to 0, this could be biasing their analysis. It would be nice to know what percentage of the data points this includes. Simply excluding such regions may be a better approach to avoid bias as non-parametric tend to approximate medians rather than means so adding enough 0s could be enough to skew interpretation.

In general we would agree with the reviewer's concern. However, in this case the impact of this transformation is negligible as the 25kb chromosome-wise ICE normalized datasets used for the non-parametric selection of top-scoring interactions do not have any invalid values (NA, NaN or Infinite). Instead, the 25kb Genome-wide ICE normalized datasets have a total of only 3 invalid values in chr2L and 2 invalid values in chr3L. Furthermore, in the results we showed about the non-parametric selection of top-scoring interactions and their subsequent clustering, in the Genome-wide ICE normalized datasets we summarized results up to 3Mb distance (Supplementary Fig. 3 a-b). In this distance range only 1 single point in chr2L contact matrix had one NA value replaced with zero.

Thus, in the methods section we explicitly stated that the "NAs, NaNs and infinite values" resulting from the normalization procedures were set 0 for the sake of transparency and for providing every technical details of the methods adopted, but in the case of the results presented and discussed in the manuscript the number of data points affected is negligible.

4) Their polymer modeling does not seem to consider PCR/sequencing biases or the idea of chromosome territories (though it does consider pairing). I'm not sure how much weight these simulations actually contribute to their interpretations.

As discussed more extensively in the reply to the first comment by Reviewer #1 above, in the revised version of our manuscript we have revised and extended the polymer folding simulations. In particular we decided to adopt a more sophisticated polymer folding simulations method incorporating the local epigenetic state of chromatin (Ghosh and Jost, PLoS Comput. Biol. 2018). This model also allowed us to consider parameters concerning the frequency of "tight" pairing between homologous chromosomes and their different prevalence in association to distinct chromatin states, based on the results reported by (Abed *et al.*, bioRxiv <https://doi.org/10.1101/443887>) on haplotype resolved Hi-C data.

Adopting a solution that is a common approach in this field (Marti-Renom and Mirny, PLoS Computat. Biol. 2011), the polymer folding simulations have been used to generate matrices of pairwise interaction frequencies. These would be comparable to contact frequencies in Hi-C data that have already been normalized, i.e. after removing PCR and other sequencing-specific biases (e.g. GC content or mappability). As such, the contact matrices derived from polymer folding simulations are not equivalent to raw Hi-C read counts (where the biases would still be present), but they are instead designed to obtain patterns directly comparable to normalized data. Thus, the PCR and sequencing-specific biases are not simulated in the polymer folding model.

Finally, for what concerns chromosome territories, these are actually implicitly accounted for in our polymer folding model. Indeed, the starting state of the simulations is from compacted chromosomes, i.e. as they would be after a mitotic cell division, then chromosomes unfold over time during the simulation. As discussed more in details in (Rosa and Everaers, PLoS Comput. Biol. 2008, and Ghosh and Jost, PLoS Comput. Biol. 2018), the dynamical properties of the polymer in terms of topological constraints and entanglements are expected to preserve the initial separation between chromosomes, thus recapitulating the properties of territories.

More details about the polymer folding simulation model have been added to the revised manuscript, in particular in the methods section that has been revised as follow:

"We modelled 4 chromosomes (2 chromosomes 3R and 2 chromosomes X) by semi-flexible self-avoiding block copolymers as described in Ghosh and Jost 2018⁴⁰. Each chromosome consisted of $N=3,200$ beads of 10kbp, each of size b . The 4 polymers moved within a cubic box of size $L_x \times L_y \times L_z$ under periodic boundary conditions in the x - y plane and rigid wall condition in the z direction to mimic the nuclear membrane.

*To each 10kbp bead i , we assigned an epigenetic state e_i based on the classification given by Filion *et al.*⁵⁸ (active, PcG, inactive and heterochromatin) where, for simplification, we merged the two active states into one unique class (see below). For chromosome 3R, to the 2790 beads that correspond to mappable regions, we added 410 heterochromatin beads to model parts of the centromeric and pericentromeric regions. Similarly, for chromosome X, 958 heterochromatin beads were inserted at the end of the 2242-long chain. The energy of a given configuration was then given by:*

$$H = \sum_{chr=1}^4 \frac{k}{2} \sum_{i=1}^{N-1} (1 - \cos \theta_{i_{chr}}) + \varepsilon \sum_{chr, chr', i, j} \delta_{i_{chr}, j_{chr'}} \Delta_{e_{i_{chr}}, e_{j_{chr}'}}$$

With k the bending rigidity, θ_i the angle between the bond vectors i_{chr} and i_{chr+1} of chromosome chr , $\varepsilon (<0)$ the contact energy between beads of the same epigenetic state, $\delta_{i_{chr}, j_{chr'}} = 1$ if beads i from chromosome chr and j from chr' occupy nearest-neighbour sites ($=0$ otherwise) and $\Delta_{e_{i_{chr}}, e_{j_{chr}'}} = 1$ if their epigenetic state is equal ($=0$ otherwise).

*Homologous pairing (between the two copies of chromosome 3R and, in the female-like situation, between the two copies of chromosome X) was accounted for by first initially forcing the homologous polymers to occupy the same position in the simulation box and then, in the rest of the simulation, by forcing some homologous beads to stay in contact. We randomly selected these forced beads so that the proportion of homologous loci in close contact, for every epigenetic classes, is the one observed by Abed *et al.*⁵⁸ using Hi-C (~100% for active, ~75% for PcG, ~50% for inactive and ~50% for heterochromatin). In*

the male-like situation (unpaired Xs), the two X copies started from different locations and no forcing was imposed. In this case, a global pairing, putatively driven by epigenetic interactions, was never observed.

*The dynamics of the chains followed a simple kinetic Monte-Carlo scheme with local moves using a Metropolis criterion applied to H . The values of $k(=1.5kT)$, $b(=105\text{ nm})$, $\varepsilon(=-0.1kT)$, $L_x=L_y(=2\mu\text{m})$ and $L_z(=4\mu\text{m})$ were fixed using the coarse-graining and time-mapping strategies developed in Ghosh and Jost⁴⁰ for a 10-nm fiber model and a volumic density= 0.009 bp/nm^3 typical of *Drosophila* nuclei. In both situations (female-like and male-like), 300 independent trajectories were simulated starting from random, compact, knot-free initial configurations with chromosomes aligned in a Rab1-like manner (all centromeric regions at the bottom of the simulation box). Each trajectory represented ~ 13 hours of real time. Note that our simulations is able to catch the formation of chromosome territories, compartments and TADs as well as homologous pairing. Predicted Hi-C data were generated by sampling 10 configurations per trajectory every 20 minutes after 10 hours (the typical time after the last mitosis for late embryos), two monomers being in 'contact' if their relative distance was less than 160nm."*

5) The fact that they don't observe a consistent decay in S2 cells (the other cell line tested clone 8 is thought to be essentially diploid) is problematic.

"We scored BG3-c2, Cl.8, D20-c2, D20-c5, D4-c1, L1, S1, W2, and D8 cell lines as minimally diploid, and S2-DRSC, S2R+, S3, Sg4, Kc167, D16-c3, and D17-c3 cell lines as minimally tetraploid. Our results for D9 and mbn2 cell line ploidy were inconclusive, due to the presence of multiple regions of relative read densities that were not ratios of whole numbers")

This shows that simply using methods that are not biased (in theory) for copy number may not be the best strategy when there are multiple X-chromosomes present (more noise) or there may be an issue with their imputation method for NAs/infinities.

For the sake of clarity, it may be worth remarking the context from which the sentence quoted in the reviewer's comment is taken from. This is a quote from a publication by (Lee et al., Genome Biology 2014), which is cited in our manuscript bibliography as well:

"We scored BG3-c2, Cl.8, D20-c2, D20-c5, D4-c1, L1, S1, W2, and D8 cell lines as minimally diploid, and S2-DRSC, S2R+, S3, Sg4, Kc167, D16-c3, and D17-c3 cell lines as minimally tetraploid. Our results for D9 and mbn2 cell line ploidy were inconclusive, due to the presence of multiple regions of relative read densities that were not ratios of whole numbers"

This sentence in the article by Lee *et al.* is referred to the ploidy estimated based on the relative distribution of DNA-seq reads coverage across multiple *Drosophila* cell lines. As the authors themselves explain, this is merely a relative estimation of ploidy as "the mean peak of DNA-seq read count density" is set to ploidy equal to "1" (see also Figure 1 in Lee et al., Genome Biology 2014). The ratios in DNA-seq reads coverage across the genome allowed the authors to determine the minimal ploidy but not an absolute ploidy. However, in the same study, by relying on additional experimental data including mitotic chromosomes spreads, the authors confirm that S1, Kc167, S2-DRSC, S2R+, S3 and D20-c5 were tetraploid. This is also in line with previous evidence and data collected by the *Drosophila* Genomics Resource Center database as well (<https://dgrc.bio.indiana.edu>).

However, the source of "noise" in S2 cells Hi-C interaction profiles is not just the mostly tetraploid karyotype, but more importantly the presence of multiple copy number variations (CNV) on top of the tetraploid background. Independent evidences in literature report indeed a very fragmented CNV profile which is also variable across different clones of S2 cells (Lee et al., Genome Biology 2014; Zhang et al., PLoS biology 2010). All of these CNV variations will have unpredictable effect on the interaction profiles as their relative positioning is not completely characterized and some of

them may also be translocated, as shown also by the original article from which we derived the Hi-C data for S2 cells (see also Supplementary Figure S7 by Ramirez et al., Mol Cell 2015).

To avoid misunderstandings on this point, we rephrased the text as follows:

"Note that this male-specific interaction decay difference is not consistent in the S2 male cell line⁸ (data not shown). This is not surprising as S2 cells are notoriously affected by several copy number and structural variations^{34,35}. These anomalies are bound to confound the genome-wide estimation of Hi-C interactions decay with distance. Furthermore, the tetraploid status of these cells, which carry two X chromosomes, might alter the general contact frequency decay behaviour."

For the sake of transparency we are reporting here the interaction decay profile for S2 cells where we can see that chrX still has some less negative slope coefficient when compared to 2R and 3R, but this result is not consistent against all autosome arms. To avoid confusion, we are currently not including this figure in the revised manuscript.

Collectively, these data in literature suggest that fine level variations in the local interaction profile, such as a change in the slope of interaction decay with distance, may be confounded by the multiple CNVs present in S2 cells, thus it is not surprising that we failed to detect the chrX specific slope difference in S2 Hi-C data. Clone 8 which is diploid and, more importantly, without major CNVs (Lee et al., Genome Biology 2014) (Drosophila Genomics Resource Center database - <https://dgrc.bio.indiana.edu>), is a more reliable model in terms of yielding a Hi-C interaction profile in line with normal diploid sex sorted male embryos data.

While the interaction decay profile may be noisy in S2 cells for the reasons discussed above, they are still a reliable model when estimating interactions differences at specific genomic positions. This was the case of the original article by (Ramirez et al., Mol. Cell 2015) that showed how interactions at MSL entry sites are present in both male and female cell lines, as well as interaction differences between sexes around roX1 and roX2 genes based on 4C data.

Finally, for what concerns the reviewer's doubt about the "imputation method for NAs/infinities", this has already been discussed and clarified in the previous reply to comment number 3 by the same reviewer (see above).

6) Having just one X-chromosome is going to intrinsically be "higher resolution" for making inferences for that particular chromosome as the issue of not being able to differentiate trans homologous chromosomal interactions does not come into play though the issue of likely having a higher percentage of trans contacts should also be taken into account.

We should remark that when dealing with Hi-C data "higher resolution" is generally achieved by a combination of:

- restriction enzyme recognizing a short sequence, thus cutting the genome with high frequency
- high coverage, in term of high sequencing depth
- summarizing contact frequencies at small genomic bin size

See for example a couple of articles achieving single Kb resolution in Hi-C data for mammalian genomes (Bonev et al., Cell 2017; Rao et al., Cell 2014) by adopting high sequencing coverage and small bin size ($\leq 1\text{Kb}$). See also (Pal et al, Biophys Rev. 2019) for a review covering this specific topic.

Instead, in this reviewer's comment we understand that "higher resolution" is referring to the ability to discriminate real intra-chromosomal (*cis*-) interactions from inter-chromosomal (*trans*-) interactions between homologous chromosomes.

We have considered the potential confounding effect of real *cis*-interactions vs *trans*-homologous interactions adopting specific analyses:

- as discussed above, and more extensively in the reply to comment #1 of Reviewer #1, we have revised and extended the polymer folding simulations. The polymer folding model now incorporates chromatin states and a more refined estimation of parameters concerning the prevalence of pairing between homologous chromosomes and its different distribution across chromatin states. These parameters have been estimated based on the most recent data in (Abed *et al.*, bioRxiv <https://doi.org/10.1101/443887>).
- The revised polymer folding model was used to disentangle the contribution of *trans*-homologous interactions between paired homologous (and their lack for male chrX) to the global interaction slope decay Supplementary Fig. 4), and the differential clustering of "top scoring" interactions (Supplementary Fig. 3f).
- The relatively higher fraction of *trans*-interactions towards other non-homologous chromosomes that is expected for the single copy male chrX has been also taken into account in the analyses reported in (Fig. 2d and 2e). As discussed in this context, the relatively higher fraction of *trans*-contacts originating from male chrX (Fig. 2d) may be confounded by the lack of a homologous pair. However, the relatively higher prevalence of interactions towards chrX for read pairs anchored in autosomes support the conclusion that chrX is more accessible (Fig. 2e). To make this point clearer, we have revised the text in the corresponding results section as that has been revised as follow:

"We noted that trans contacts are relatively more frequent specifically on male chrX, as confirmed both in sex-sorted embryos and cell lines datasets (Fig. 2d, and Supplementary Fig. 3d). This is compatible with a globally more accessible male chrX. However, we reasoned that this might also be due to a technicality of Hi-C data analysis, as trans contacts between homologous chromosomes are indistinguishable from intra-chromosomal contacts. As male chrX is the only chromosome without a homologous one in our dataset, the relatively higher number of trans contacts might just be due to the absence of confounding contacts between homologues. To rule out this possibility, we examined the distribution of trans contacts across chromosomes and verified that in sex-sorted embryos trans contacts originating from each single male autosome are specifically enriched in chrX (Fig. 2e). This pattern is not compatible with a confounding effect due to the single copy of chrX and can only be explained by chrX indeed engaging in more trans contacts with all other chromosomes. Conversely this pattern is not observed in female sex-sorted embryos (Fig. 2e)"

7) Instead of down sampling they should try to up-sample the male reads for the X so that there is less "noise".

Unfortunately, we respectfully disagree with this suggestion.

The random down-sampling (sub-sampling without replacement) of aligned Hi-C sequencing reads recapitulates what would be observed if less reads are sequenced.

Instead, “up-sampling” reads by definition can’t recapitulate the effect of higher sequencing depth. Indeed this “up-sampling” could be achieved either 1) by applying sampling with replacement on the observed (real) sequencing reads; or 2) by randomly placing new reads on the genome.

The first approach would result in sampling multiple times some reads. This would be actually simulating the PCR duplicates artifacts that are normally removed in Hi-C data preprocessing. Thus, this approach would create a spurious signal that normally is filtered out of the Hi-C contact matrix.

The second approach instead would create completely random Hi-C signal, thus not recapitulating the effect of higher sequencing depth, but adding instead noise to the data.

Thus, none of the possible “up-sampling” procedures can achieve the goal of obtaining “less noisy” Hi-C data, in the terms proposed by this reviewer’s comment.

8) What is the rationale for leaving out the Het regions or not using dm6?

We acknowledge that dm6 would be the latest reference genome assembly for *Drosophila melanogaster*. It may also be worth remarking that, due to a change in the reference genome assemblies naming convention, the dm6 is actually the genome build following the dm3 (*i.e.* there are no dm4 or dm5 builds).

In our analyses we used dm3 as reference for the following reasons:

- to avoid any potential comparability issues with the 2 previous articles focusing on dosage compensation and Hi-C data (Ramirez et al., Mol Cell 2015; Schauer et al., EMBO Rep. 2017)
- to be able to incorporate results from previous literature in the field of *Drosophila* dosage compensation, which is mostly based on dm3, without the need to apply strategies to convert genomic coordinates across different builds, such as the "liftover" tool by UCSC genome browser (<http://genome.ucsc.edu/cgi-bin/hgLiftOver>). E.g. previous publications with MSL entry sites (Alekseyenko et al., Cell 2008; Straub et al., PLoS Genet 2008; Ramirez et al., Mol Cell 2015), or CLAMP binding sites (Sorucu et al., Genes & Development 2013) are based on dm3 and these data are used in our manuscript.
- likewise, to be able to compare results with other sources of data, such as modENCODE, that primarily adopted dm3, without the need to apply genomics coordinate conversions tools such as liftover. While these tools are generally reliable, they may result in losing some details in the conversion.
- It is a common best practice to freeze the reference genomic annotations used for a project. As we started using dm3 for the best comparability with previous literature, as described above, thus we preserved the dm3 as reference also later.

Finally, it must be noted that the availability of dm6 does not mean that dm3 should be considered outdated. In fact, other very recent articles in the field rely as well on dm3 as reference genome, such as for example (Ogiyama et al., Mol Cell 2018) or (Lam et al., Genes & Development 2019).

For what concerns the choice of leaving out from our analyses the "Het" portions of chromosomes 2, 3 and X, as well as the entire chr4 and chrY, this was as well motivated by the choice to be in line with previous literature (Ramirez et al., Mol Cell 2015). The rationale behind this solution is that chr4, chrY and all of the "Het" portions of other chromosomes are 1) very short in size (with the latest genome assembly containing only 1.3Mb long sequence for chr4 and 3.7Mb sequence for the Y chromosome), thus not comparable to chrX and other autosome arms when assessing metrics like interaction decay with distance; 2) they are known to be

characterized by a strongly heterochromatic epigenetic status (Kharchenko et al., Nature 2011; Riddle et al., Genome Research 2011). Thus, they may not be representative of the average autosomal genomic regions, i.e. subjected to normal gene regulation mechanisms, to be used as reference comparison for dosage compensated chrX (see also Johansson et al. Molecular and Cellular Biology 2012). To explicitly explain this choice, we have revised the text in the methods section as follow:

"[...] the sequencing reads were aligned to the dm3 genome build considering only chromosomes X, 2 and 3. Chromosome 4, Y and the heterochromatic portions (named with suffix "Het") were left out. The rationale behind this solution is that chr4, chrY and all of the "Het" portions of other chromosomes are very short in size, thus not comparable to chrX and other autosome arms when assessing metrics like interaction decay with distance. Moreover, they are known to be characterized by a strongly heterochromatic epigenetic status^{52,54}. Thus, they may not be representative of the average autosomal genomic regions to be used as reference comparison for dosage compensated chrX (see also Johansson et al. ⁵⁵)."

REVIEWERS' COMMENTS:

Reviewer #1 (Remarks to the Author):

All my previous concerns have been addressed.

Reviewer #2 (Remarks to the Author):

The authors did great job in the revision and fully addressed my previous comments and suggestions. The manuscript has been significantly improved. I don't have further comments.

Reviewer #3 (Remarks to the Author):

The authors have done an exceptionally strong job revising their manuscript. They updated their polymer model and incorporated new data that allows cis- and trans-interactions to be distinguished.

This is a very strong addition to the field and should definitely be published in Nature Communications.